# Impaired spatial learning and suppression of sharp wave ripples by cholinergic activation at the goal location

**Przemyslaw Jarzebowski†, Clara S Tang†, Ole Paulsen, Y Audrey Hay***

Department of Physiology, Development and Neuroscience, Physiological Laboratory, Cambridge, United Kingdom

**Abstract** The hippocampus plays a central role in long-term memory formation, and different hippocampal network states are thought to have different functions in this process. These network states are controlled by neuromodulatory inputs, including the cholinergic input from the medial septum. Here, we used optogenetic stimulation of septal cholinergic neurons to understand how cholinergic activity affects different stages of spatial memory formation in a reward-based navigation task in mice. We found that optogenetic stimulation of septal cholinergic neurons (1) impaired memory formation when activated at goal location but not during navigation, (2) reduced sharp wave ripple (SWR) incidence at goal location, and (3) reduced SWR incidence and enhanced theta-gamma oscillations during sleep. These results underscore the importance of appropriate timing of cholinergic input in long-term memory formation, which might help explain the limited success of cholinesterase inhibitor drugs in treating memory impairment in Alzheimer's disease.

## Introduction

The role of the neuromodulator acetylcholine (ACh) in learning and memory is debated. On one hand, degeneration of cholinergic neurons and a low tone of ACh correlate with memory impairment in humans and rodents (*Bartus, 2000*; *Berger-Sweeney et al., 2001*; *Hasselmo and Sarter, 2011*) and drugs blocking ACh degradation ameliorate memory impairment in Alzheimer's disease (AD) (*Ehret and Chamberlin, 2015*). Moreover, extensive lesions of the medial septum (MS), where the cholinergic neurons projecting to the hippocampus are located, produce learning deficits (*Hepler et al., 1985*). On the other hand, selective lesions of MS cholinergic neurons, which account for ~5% of neurons in the MS, have relatively little impact on learning and memory (for review, see *Hasselmo and Sarter, 2011*; *Solari and Hangya, 2018*). A reason for these apparently conflicting results may lie in the fact that memory formation is a dynamic process during which ACh levels vary (*Fadda et al., 2000*), a property lesioning or pharmacological studies cannot directly address.

In an influential model, *Buzsáki, 1989* suggested that long-term memory forms in two stages: first, neuronal activity produces a labile trace, then a delayed potentiation of the same synapses forms a long-lasting memory trace. He suggested that the two stages are associated with different network states of the hippocampus. In rodents, phase-amplitude-coupled theta (5–12 Hz)-gamma (30–100 Hz) oscillations (*Csicsvari et al., 2003*) occur during exploratory behaviors, while large-amplitude sharp waves combined with high-frequency ripples (sharp wave ripples [SWRs]) (*O'Keefe and Nadel, 1978*; *Csicsvari et al., 2000*) occur during immobility and slow-wave sleep. These two network states are mutually exclusive (*O'Keefe and Nadel, 1978*; *Buzsáki, 1986*; *Csicsvari et al., 2000*), and it was suggested that memory encoding is associated with theta/gamma oscillations while memory consolidation relies on SWR activity (*O'Neill et al., 2010*; *Colgin, 2013*).

The local release of ACh controls hippocampal network states. In hippocampal CA3, cholinergic activation ex vivo induces a slow gamma rhythm primarily by activating M1 muscarinic receptors

**\*For correspondence:**
ah831@cam.ac.uk

†These authors contributed equally to this work

**Competing interests:** The authors declare that no competing interests exist.

(*Fisahn et al., 1998*; *Betterton et al., 2017*), while in the CA1, cholinergic activation in vivo promotes theta/gamma oscillations and suppresses ripple oscillations through the activation of M2/M4 muscarinic receptors (*Vandecasteele et al., 2014*; *Zhou et al., 2019*; *Ma et al., 2020*). This suggests that regulation of cholinergic tone allows the switching between online attentive processing (theta/gamma oscillations) and offline memory consolidation (SWRs) as described in the two-stage model of memory trace formation (*Buzsáki, 1989*). Evidence from microdialysis and electrophysiology experiments shows a high cholinergic tone during exploration, promoting theta activity, and a lower cholinergic tone during subsequent rest, permitting SWRs (*Fadda et al., 2000*; *Giovannini et al., 2001*; *Fadel, 2011*). Disruption of cholinergic activity at different stages of learning and memory impairs performance in memory tasks (for review, see *Hasselmo and Sarter, 2011*; *Solari and Hangya, 2018*). However, the differential effects of ACh in distinct phases of memory formation are not well understood.

To clarify the function of the MS cholinergic system during hippocampus-dependent learning and memory, we investigated the behavioral phase-specific effects of optogenetic cholinergic stimulation in the appetitive Y-maze long-term memory task, a simple reward-based spatial memory task with two distinct behavioral phases: one of navigation toward a reward and another after arriving in the goal area (*Bannerman et al., 2012*). We show that stimulation of cholinergic neurons does not affect learning when applied during navigation toward a reward but impairs learning when applied at the goal location. Our simultaneous recordings of hippocampal local field potential (LFP) indicate that impaired memory was related to disruption of awake SWRs, supporting the two-stage model of memory trace formation (*Buzsáki, 1989*). We also show that activation of MS cholinergic neurons promotes a switch from ripple activity to enhanced theta/gamma oscillations in the hippocampus of naturally sleeping mice.

## Results

### Functional expression of ChR2 in cholinergic neurons

We aimed to investigate the effects of cholinergic modulation on hippocampal oscillations and performance in a spatial navigation task. To this end, we optogenetically controlled the activity of cholinergic neurons using ChAT-Ai32 crossbred mice that expressed enhanced YFP-tagged channelrhodopsin-2 (ChR2-eYFP) under the control of the choline-acetyl transferase (ChAT) promoter. We first confirmed the expression of ChR2 in MS cholinergic neurons by performing double immunostaining for ChAT and YFP (*Figure 1A*). In sections sampled from two mice, 98 out of 150 ChAT+ cells counted were YFP+ (YFP+/ChAT+=65%). In independently sampled sections, 111 out of 111 YFP+ cells counted were ChAT+ (ChAT+/YFP+=100%).

We probed the functional expression of ChR2 in MS neurons by recording multi-unit activity in the MS of urethane-anesthetized mice; 473 nm light was delivered through an optic fiber implanted just above the MS while multi-unit activity was recorded from a co-assembled electrode the tip of which protruded ~200 μm further than the optic fiber tip. Local light delivery (50 ms pulses at 10 Hz) resulted in an increase in multi-unit activity (baseline spike frequency: 7.9 ± 2.8 Hz vs. spike frequency during light delivery: 22.7 ± 7.3 Hz, two-tailed Wilcoxon matched pair signed-rank test: p=0.03; n = 6 recordings from two mice; *Figure 1B*), confirming our ability to increase neuronal activity in the MS using optogenetics.

### Activation of septal cholinergic neurons in the goal zone slows place learning

We investigated the effect of cholinergic activation during different phases of the appetitively motivated Y-maze task, a hippocampus-dependent task commonly used to study long-term spatial memory (*Bannerman et al., 2012*; *Shipton et al., 2014*). Mice had to learn to find a food reward on an elevated three-arm maze that remained at a fixed location in relation to visual cues in the room, while the mice pseudo-randomly started from one of the other two arms. Because short-term memory errors caused by re-entry during a single trial have previously been shown to interfere with the acquisition of this spatial long-term memory task (*Schmitt et al., 2003*), mice were only allowed to make a single choice of arm in each trial (*Figure 2A*). Previous studies have reported sharp wave ripples at the reward location of different spatial navigation tasks (*O'Neill et al., 2006*;

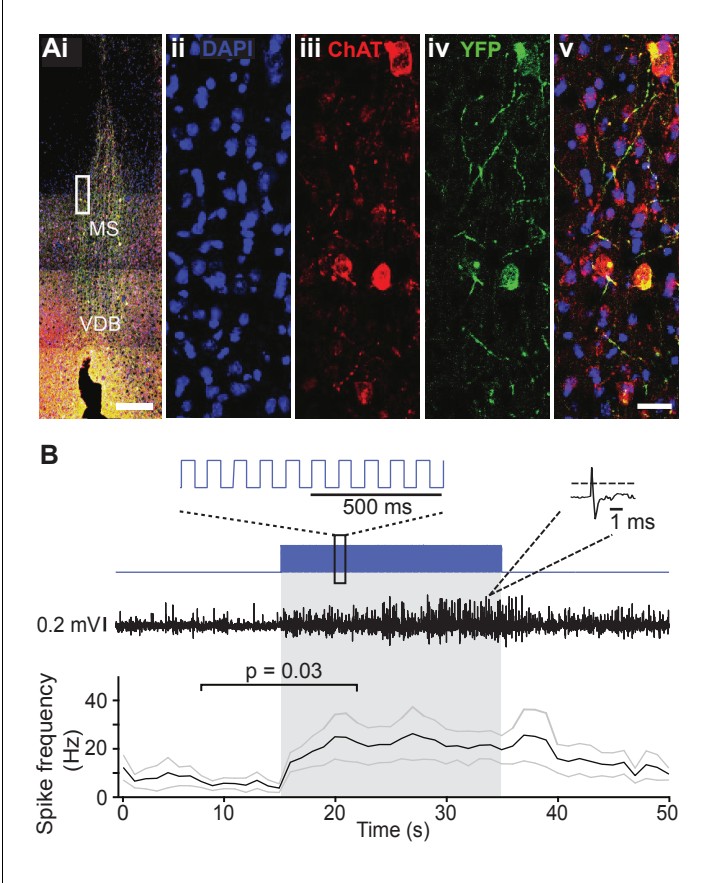

**Figure 1.** Choline acetyl transferase (ChAT)-Ai32 mice express enhanced YFP-tagged channelrhodopsin-2 (ChR2-eYFP) selectively in cholinergic cells. (**Ai**) Overlay of DAPI, ChAT, and eYFP-positive immunostaining in a coronal section of the medial septum (MS) in a ChAT-Ai32 mouse. Scale bar 500 μm. VDB, ventral diagonal band. (**Aii-v**) Higher magnification of the MS (rectangle in **Ai**), triple immunostaining of DAPI (blue, **ii**), ChAT (red, **iii**), and eYFP (green, **iv**), showing their colocalization (overlay, **v**). Scale bar 50 μm. (**B**) Sample trace of multi-unit recording from the MS in a ChAT-Ai32 mouse. Top: the stimulation protocol (blue) beginning at 15 s. Inset shows a section of the 50-ms-long square stimulation pulses at 10 Hz. Middle: an example recording trace; inset shows an example unit recorded. Bottom: mean spike frequency (n = 6). *p=0.03, two-tailed paired Wilcoxon signed-rank test. Gray lines represent mean ± SEM.

*Dupret et al., 2010*). Based on these studies, we defined a navigation phase corresponding to the arms of the maze except for the distal ends (20 cm from the edge), which we considered as goal zones. Cholinergic activation was achieved by light stimulation (473 nm, 25 mW, 50-ms-long pulses at 10 Hz) delivered via an optic fiber implanted in the MS of ChAT-Ai32 mice. ChAT-Ai32 mice were split into four groups to test four experimental conditions: (i) no stimulation (n = 13), (ii) optogenetic stimulation during navigation – from the start of the trial until they reached the goal zone (n = 9), (iii) optogenetic stimulation throughout the maze (n = 9), and (iv) optogenetic stimulation in the goal zone only – from the entry of the goal zone until the mice were removed from the maze either after they had eaten the food or they had reached the empty food well (n = 15; *Figure 2B*).

Each mouse received 10 trials per day for 6–10 consecutive days, and we set a learning criterion of ≥80% rewarded trials in a day. Mice from all four groups of ChAT-Ai32 mice learned the task (*Figure 2C*, *Figure 2—figure supplement 1*) but comparison between the groups revealed differences in the number of days taken to reach this criterion (one-way ANOVA on ranks $\chi^2(3)=14$, p=0.003, $BF_{10} = 20$; *Figure 2E*). Post hoc tests indicated that the ChAT-Ai32 'goal' group was delayed at learning the task compared to the ChAT-Ai32 'no stimulation' group (4.5 ± 0.3 vs. 2.9 ± 0.3 days, Dunn test with Holm-Bonferroni correction for multiple comparisons: p=0.002, $BF_{10} = 45$). Similarly, the ChAT-Ai32 'goal' group was delayed compared to the ChAT-Ai32

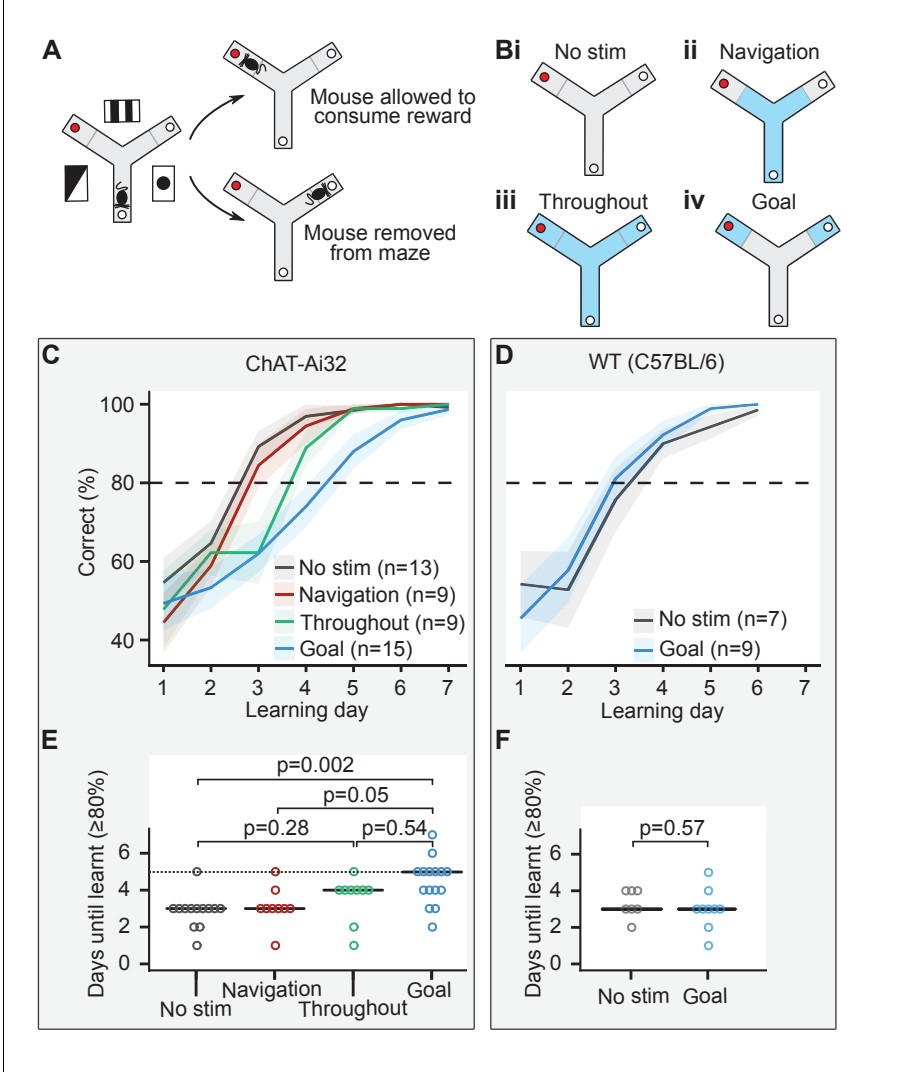

**Figure 2.** Cholinergic stimulation in the goal zone slows learning of the appetitive Y-maze task. (A) Mice were trained on an elevated three-arm maze to find a food reward (red dot) in an arm that remained at a fixed location relative to visual cues in the room. Mice were allowed to consume the reward if they chose the correct arm but were removed from the maze if they chose the incorrect arm. (B) Mice were pseudo-randomly split into four groups to test four optogenetic stimulation conditions. Blue indicates stimulation for the four conditions: (i) no stimulation, (ii) stimulation only until the goal zone was reached (gray line), (iii) stimulation throughout the maze, and (iv) stimulation only in goal zone. (C) Choline acetyl transferase (ChAT)-Ai32 mice received blocks of 10 trials each day for 7 consecutive days and the number of entries to the rewarded arm was recorded. The performance of all four groups of mice improved with time but at different rates. (D) As in (C) but for wild-type (WT) mice. (E) The number of days required for each group of the ChAT-Ai32 mice to reach the learning criterion of ≥80%. Horizontal bars indicate the median within each group. The p-values for differences between groups were calculated using post hoc Dunn tests. (F) As in (E) but for WT mice.

The online version of this article includes the following source data and figure supplement(s) for figure 2:

**Source data 1.** ChAT-Ai32 learning performance.
**Source data 2.** WT mice learning performance.
**Figure supplement 1.** Individual learning curves.
**Figure supplement 2.** Optic fiber implant placement.

'navigation' group (4.5 ± 0.3 vs. 3.1 ± 0.4 days, p=0.05, $BF_{10}$ = 4.4). Whilst the stimulation for the 'goal' group lasted longer (34 ± 1 vs. 8 ± 1 s), the duration alone cannot explain the different effects of the optogenetic stimulation. The 'throughout' group received the longest stimulation (42 ± 1 s) but presented an intermediate learning curve. Using Bayes Factor (BF) analysis, we found inconclusive evidence for the 'throughout' group to learn more slowly than the 'no stimulation' group (post hoc test for difference in means: p=0.28, test for higher mean days-to-criterion in the 'throughout' group: $BF_{10}$ = 1.6) and learn faster than the 'goal' group (post hoc test for difference in means: p=0.54, test for lower mean days-to-criterion in the 'throughout' group: $BF_{10}$ = 2.0). Therefore, the spatial location in the maze where the optogenetic stimulation took place was most likely the factor that decided the behavioral outcome. However, the MS neurons sustained an increased level of firing after the optogenetic stimulation ceased (*Figure 1B*). Therefore, we cannot exclude the possibility that this sustained activity contributes to the learning deficit in the 'goal' group.

To control for possible aversive or other non-specific effects of the illumination, we performed an additional experiment with MS-implanted wild-type (WT) mice split into two groups: no stimulation (n = 7) and light delivery in the goal zone (n = 9; *Figure 2D*). We did not observe any learning difference between the 'goal' and 'no stimulation' groups of this control WT mice cohort (goal: 3.0 ± 0.37 days; no stimulation: 3.4 ± 0.37 days; one-way ANOVA $F_{(1, 14)}$ = 0.34, p=0.57; *Figure 2F*).

We confirmed that the memory of the rewarded arm was retained by retesting the mice on the Y-maze task 1 week after the end of the acquisition period for each group of the ChAT-Ai32 mice (no stimulation: 100 ± 0%; navigation: 99 ± 1%; throughout: 100 ± 0%; goal: 94 ± 3%). After behavioral testing, implant placement and the level of eYFP expression were verified by immunohistochemistry, confirming that there were no significant differences in implant placement between the behavioral groups (*Figure 2—figure supplement 2*).

Our results show that cholinergic activation in the goal zone for as short as 50 s (95% percentile of stimulation duration) slows learning of the appetitive Y-maze task. In contrast, optogenetic stimulation during navigation or throughout the maze had no significant effect on task acquisition.

## Activation of MS cholinergic neurons in the goal zone does not significantly affect theta-gamma power but reduces the incidence of SWRs

To understand why stimulating MS cholinergic neurons in the goal zone impairs task acquisition, we performed LFP recordings from the hippocampus during task performance. We recorded CA1 field potentials during the Y-maze task in five ChAT-ChR2 and two control ChAT-GFP mice implanted with recording electrodes and an optic fiber. We used staggered wire electrodes to record the field potentials and subtracted the signal in one electrode from that in the other. This subtraction procedure cancels out synchronous changes on both electrodes, like those caused by movement artifacts, and enhances locally generated phase-reversed signals, such as theta, gamma, and ripple events. Optogenetic stimulation was applied on alternating trials when the mouse reached the goal zone, comparing the CA1 activity between the stimulated and non-stimulated trials (111 non-stimulated and 109 stimulated at the goal location rewarded trials and 56 non-stimulated and 36 stimulated at the goal location unrewarded trials). To evaluate the effects of laser (on vs. off), mouse group (ChAT-ChR2 vs. ChAT-GFP), and their interaction, while accounting for correlations between the trials for the same mouse, we used a linear mixed-effects model (see 'Materials and methods'). The cholinergic activation did not overtly affect the behavior once the mice were at the goal location: we did not detect any effect of the laser on the time the mice spent at the goal location (linear mixed-effects model, mouse group–laser interaction: $F_{(1, 78)}$=0.01, p=0.94, laser effect: $F_{(1, 78)}$=0.1, p=0.73).

The theta power (5–12 Hz) peaked in the central section of the maze where the mice ran the fastest (*Figure 3A,B*) and was reduced at the goal location in rewarded trials (*Figure 3B*, right panel). Power spectral density (PSD) from electrophysiological recordings measures the summation of periodic activity and aperiodic activity. The intensity of the aperiodic component of the PSD has a pink noise distribution (1/f) (*Donoghue et al., 2020*). Therefore, to quantify the power of theta and gamma oscillations, we measured relative peaks above the estimated aperiodic component (*Donoghue et al., 2020*, see 'Materials and methods', *Figure 3C,D*).

Both theta and slow gamma (25–80 Hz) oscillations were present at the goal location in the rewarded non-stimulated trials (theta peak present in 98 ± 2% of trials; slow gamma peak present in 86 ± 7% trials, *Figure 3C* and *Figure 3—figure supplement 1*). Light did not affect the relative theta

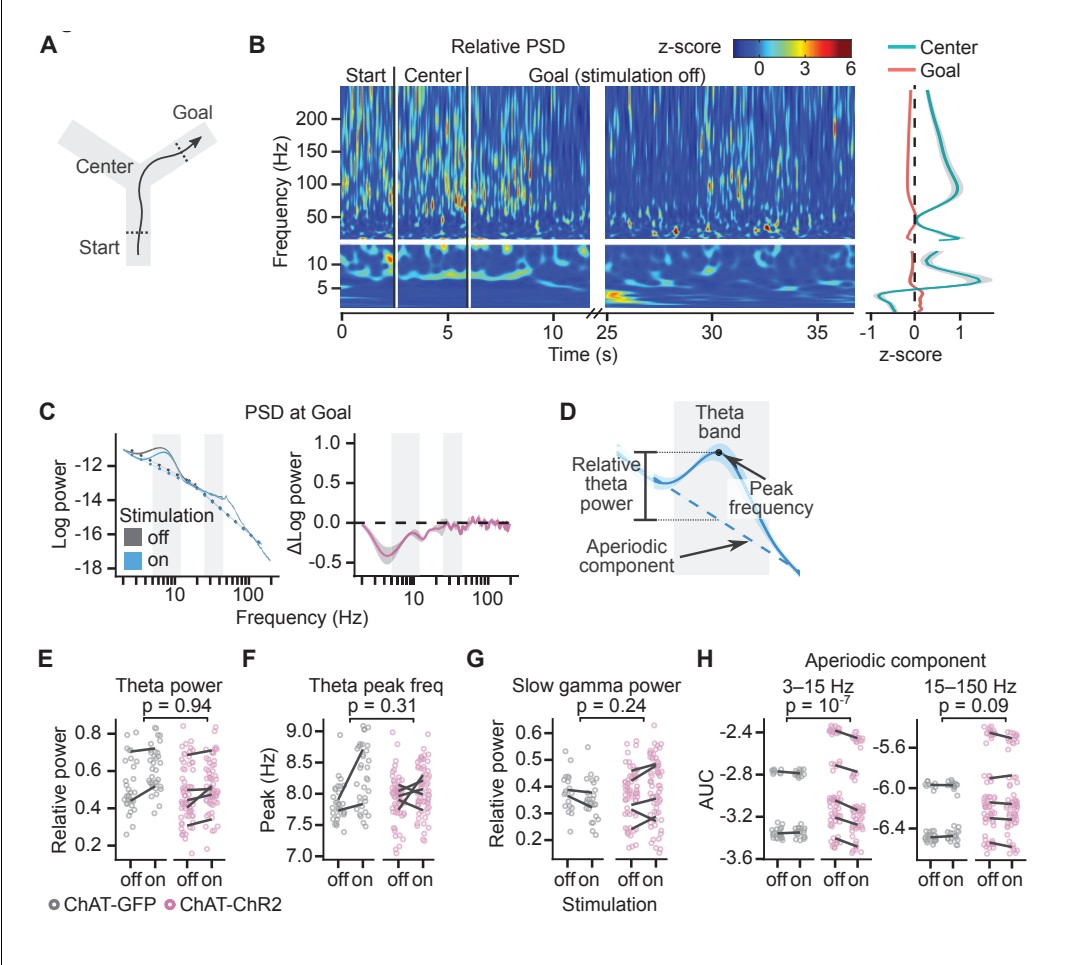

**Figure 3.** Cholinergic stimulation during the Y-maze task did not change theta-gamma oscillations. (**A**) Schematic showing the maze zones. (**B**) Spectrogram of the local field potential (LFP) recorded in a single non-stimulated trial. The values were z-scored to show relative changes in frequency. Note transient increases in high-frequency power throughout the recording and high theta power at Center. Right: mean z-score value at Center and Goal as a function of frequency. (**C**) Left: power spectral density (PSD) of the LFP recorded from a representative animal on non-stimulated and stimulated-at-Goal rewarded trials. The dashed lines show the fitted aperiodic component. Right: difference in PSD between day-averaged trials with stimulation off and on. Ribbons extend ±1 SEM of log power. Gray background marks the frequency range of theta and slow gamma bands. (**D**) PSD parameters that were assessed for the stimulation effect: relative theta power (**E**), spectral peak frequency in the theta band (**F**), slow gamma power (**G**), and the aperiodic component (**H**). The aperiodic component was fitted for two frequency ranges, 3–15 and 15–150 Hz, and compared using the area under curve (AUC). (**E–H**) Values plotted for individual trials. Lines connect means for individual animals. p-values were calculated with linear mixed-effects models for the interaction of mouse group–laser effects.

The online version of this article includes the following source data and figure supplement(s) for figure 3:

**Source data 1.** Theta power.

**Source data 2.** Theta peak frequency.

**Source data 3.** Slow gamma power.

**Source data 4.** Aperiodic component power.

**Figure supplement 1.** Power spectral density (PSD) at Goal location.

**Figure supplement 2.** Power spectral density (PSD) change of theta and slow gamma at Goal location vs. neighboring frequency band.

**Figure supplement 2—source data 1.** PSD change per frequency band.

power differently in the ChAT-GFP and ChAT-ChR2 mice (linear mixed-effects model, mouse group–laser interaction: $F_{(1, 5)}$=0.01, p=0.94, *Figure 3E*). However, theta power increased by 16 ± 3% in both mouse groups indiscriminately (laser effect: $F_{(1, 5)}$=6.9, p=0.05), suggesting another, non-specific effect of the laser on theta power. Spectral peak frequency in the theta band was not

significantly affected by the stimulation (linear mixed-effects model, mouse group–laser interaction: $F_{(1, 5)}$=1.3, p=0.31, laser effect: $F_{(1, 5)}$=3.5, p=0.12, *Figure 3F*).

To independently confirm that the stimulation did not differentially affect relative theta power in ChAT-GFP and ChAT-ChR2 mice, we looked at the difference in the PSD between day-averaged trials with the stimulation off and on (*Figure 3C* right panel, *Figure 3—figure supplement 1*). Differences for a given frequency can be caused by a change in oscillatory power, change in the aperiodic component, or by a shift of the spectral peak frequency or change of the peak's width (bandwidth). To minimize the impact of the peak frequency shift and change in bandwidth, we compared maximum changes within frequency bands that were wider than the bandwidth of the theta peak and shift in theta peak frequency. In the ChAT-ChR2 mice, the difference between the negative power change in the theta band and in the surrounding bands was not significantly different than in the ChAT-GFP mice (*Figure 3—figure supplement 2*). Hence, our results indicate that the stimulation did not affect the theta power significantly differently between the mouse groups.

Similarly, quantification of the relative slow gamma power indicated no effect of the stimulation (linear mixed-effects model, mouse group–laser interaction: $F_{(1, 6)}$=1.7, p=0.24, laser effect: $F_{(1, 6)}$=0.1, p=0.77, *Figure 3G*). This result was independently confirmed by looking at the difference in the PSD between day-averaged trials with the stimulation off and on (*Figure 3—figure supplement 2*).

The only effect of the laser that affected PSD of the ChAT-GFP and ChAT-ChR2 mice differently was a reduced aperiodic component of the PSD in the 3–15 Hz range (linear mixed-effects model, mouse group–laser interaction: $F_{(1, 135)}$=29, p=$10^{-7}$, *Figure 3H*).

Previous studies have shown that activation of septal cholinergic neurons suppresses CA1 SWRs (*Vandecasteele et al., 2014*; *Ma et al., 2020*), and we investigated whether impaired place learning was associated with changes in SWRs. To identify the SWRs, we detected ripple events in the LFP and excluded any candidate ripples that co-occurred with electromyography (EMG)-detected muscle activity. Only ripples with spectral peak frequency ≥140 Hz were identified as SWRs (*Sullivan et al., 2011*; *Figure 4A*, *Figure 4—figure supplements 1* and *2*).

The SWRs occurred at the start and the goal locations (*Figure 4B*). Mice learned over 6 days and on day 5 reached 80 ± 10% rewarded trials. We detected SWRs in significantly more rewarded than unrewarded trials (82 ± 7% of rewarded non-stimulated trials vs. 32 ± 13% of unrewarded non-stimulated trials, paired t-test on percentages per animal: p=0.02, n = 7 animals, *Figure 4C*). The difference could be due to the shorter immobility when the mice visited the non-rewarded arms: on unrewarded trials, mice spent 6.5 ± 0.5 s in the goal zone before leaving compared to 34.0 ± 1.0 s on rewarded trials. Because we detected few SWRs in the unrewarded trials, we restricted the further analysis to the rewarded trials.

We first assessed whether SWR incidence changed during learning by quantifying the incidence of SWRs in the non-stimulated rewarded trials during early and late learning (*Figure 4D*). We did not observe any significant difference between early (before day 5) and late learning (linear mixed-effects model, effect of early vs. late learning: $F_{(1, 110)}$=0.3, p=0.58, *Figure 4D*). However, optogenetic stimulation had a significantly different effect in the ChAT-GFP and the ChAT-ChR2 mice (log-linear mixed-effects model, mouse group–laser interaction, $F_{(1, 42)}$ = 4.5, p=0.04, *Figure 4E*), whose SWR incidence at the goal location was reduced by 52 ± 7% from 0.06 ± 0.01 to 0.03 ± 0.01 Hz (post hoc test: $t_{(44)}$ = 4.2, p=0.001, *Figure 4E*). Spectral peak frequency of SWRs was not affected by the stimulation (frequency: 168 ± 2 Hz; linear mixed-effects model for non-stimulated trials, mouse group–laser interaction: $F_{(1, 3.6)}$=0.02, p=0.88, *Figure 4—figure supplement 2A*), nor was the SWR duration (duration: 37 ± 1 ms; log-linear mixed-effects model for non-stimulated trials, mouse group–laser interaction: $F_{(1, 148)}$=0.1, p=0.76, *Figure 4—figure supplement 2B*).

Overall, these results show that optogenetic stimulation of MS cholinergic neurons reduced ripple incidence in the CA1 in rewarded trials but did not cause a detectable change in theta-gamma power. Hence, this result suggests that the reduced SWR incidence is a mechanism relevant for the memory impairment induced by cholinergic stimulation in this task.

## MS cholinergic neuron stimulation reduces SWRs and increases theta and slow gamma activity in sleeping animals

Because cholinergic input has been implicated in theta activity in the hippocampus (*Buzsáki, 2002*), we were surprised that we could not detect any effect on theta-gamma oscillations by cholinergic

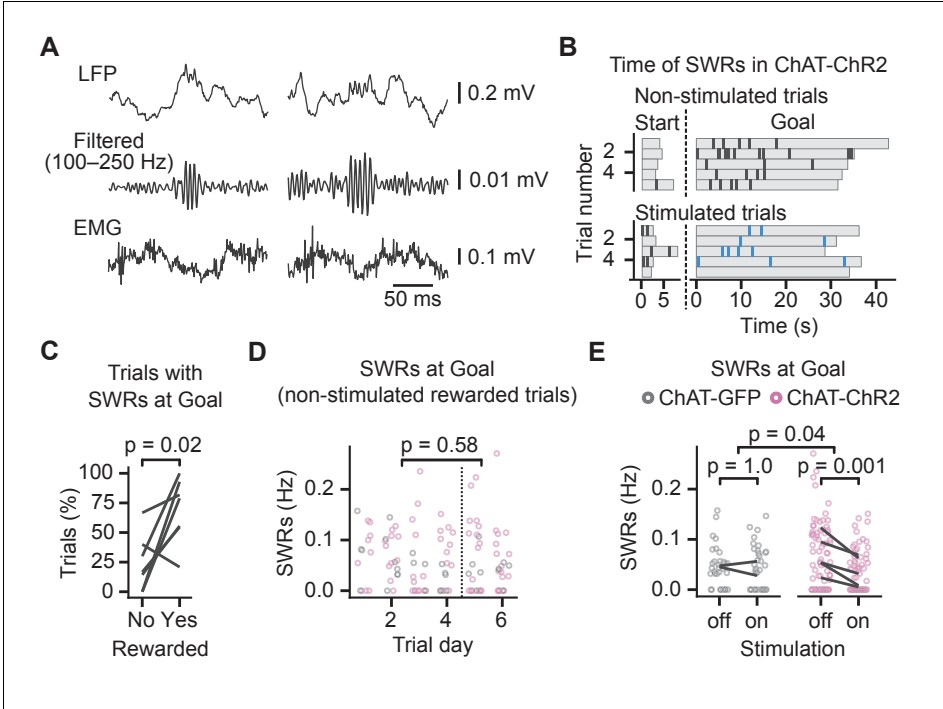

**Figure 4.** Cholinergic stimulation during the Y-maze task reduced incidence of sharp wave ripples (SWRs). (**A**) Example SWRs recorded at the Goal location: local field potential (LFP) traces (top), the same traces after 100–250 Hz bandpass filtering (middle), and simultaneously recorded electromyography (EMG) (bottom). (**B**) Time of SWRs recorded from a representative mouse over multiple trials the same day. Time measured relative to the trial start and arrival at Goal. Stimulated and non-stimulated trials are grouped for clarity. (**C**) Percentage of trials with SWRs at Goal compared between unrewarded and rewarded non-stimulated trials. One line per animal shown. p-value calculated with paired t-test. (**D**) SWR incidence as a function of trial day. Data shown for choline acetyl transferase (ChAT)-GFP and ChAT-ChR2 mice together, values plotted for individual non-stimulated rewarded trials. Dashed vertical line separates early and late trials. p-value calculated with linear mixed-effects model for the effects of early vs. late trials. (**E**) Effect of cholinergic stimulation on SWR incidence at Goal location in rewarded trials. Data shown for ChAT-GFP and ChAT-ChR2 mice, values plotted for individual trials. Lines connect means per animal. p-values calculated with linear mixed-effects model for the interaction of mouse group–laser effects; groups were compared with post hoc test on least-square means.

The online version of this article includes the following source data and figure supplement(s) for figure 4:

**Source data 1.** Time of SWRs.

**Source data 2.** Trials with SWRs at Goal.

**Source data 3.** SWR incidence at Goal over learning.

**Source data 4.** SWR incidence at Goal in stimulated vs non-stimulated trials.

**Figure supplement 1.** Sharp wave ripples (SWRs) recorded at Goal location.

**Figure supplement 2.** Spectral peak frequency of ripples and duration of sharp wave ripples (SWRs) at Goal location.

**Figure supplement 2—source data 1.** Ripple spectral peak frequency and duration.

stimulation at the goal location. However, there are at least two distinct forms of theta oscillations in the hippocampus, only one of which is dependent on cholinergic receptors and can be observed during sleep (*Kramis et al., 1975*). Also, the reduction of SWR incidence of 52 ± 7% at the goal location was smaller than the 92% median suppression reported during free behavior (*Vandecasteele et al., 2014*), which could be due to a smaller effect of ACh at the reward location or an already high level occluding the effect of the optogenetic stimulation. We therefore compared the effects of cholinergic stimulation during task performance with the effects of cholinergic stimulation during sleep when cholinergic tone is at the lowest.

We recorded LFP signal while the mice slept in a cage, to which they had been familiarized over the 2 previous days and alternated periods without optogenetic stimulation (60–120 s) and periods

with optogenetic stimulation (30 s). We compared the signal in the 30-s-long epochs preceding the stimulation with the 30-s-long epochs during the stimulation without a distinction between SWS and REM sleep. Only epochs during which the mouse was asleep for their full duration were used for the analysis (n = 369 epochs from 10 animals, IQR of 5–16 epochs in succession without interrupted sleep).

Optogenetic stimulation reduced the SWR incidence throughout the stimulation in ChAT-ChR2 mice but not in ChAT-GFP mice (*Figure 5A–C*). SWR incidence in ChAT-ChR2 mice was reduced from 0.21 ± 0.01 to 0.03 ± 0.01 Hz (85 ± 3% reduction, linear mixed-effects model, mouse group–laser interaction: $F_{(1, 22)}=47$, $p=10^{-6}$, n = 369 epochs from 10 animals; post hoc test for laser effect in ChAT-ChR2: $t_{(86)} = 9.7$, $p=10^{-14}$, *Figure 5C*). The stimulation did not change the spectral peak frequency of the SWRs of 168 ± 1 Hz (mouse group–laser interaction: $F_{(1, 262)}=0.51$, $p=0.48$, *Figure 5—figure supplement 1A*), nor ripple duration of 38 ± 0.3 ms (log-linear mixed-effects model, mouse group–laser interaction: $F_{(1, 28)}=0.9$, $p=0.35$, *Figure 5—figure supplement 1B*). Hence, our results confirm that optogenetic activation of MS cholinergic neurons almost completely suppresses SWRs during sleep in the CA1.

We next investigated the effect of optogenetic stimulation of MS cholinergic neurons on theta-gamma activity in the sleeping mouse (*Figure 6A*). We determined the PSD for frequencies ranging between 1 and 200 Hz in control condition and during stimulation. We observed a reduction of the PSD across the full frequency range upon light stimulation (*Figure 6B–D*, *Figure 6—figure supplement 1*), as reported previously in freely behaving mice (*Vandecasteele et al., 2014*).

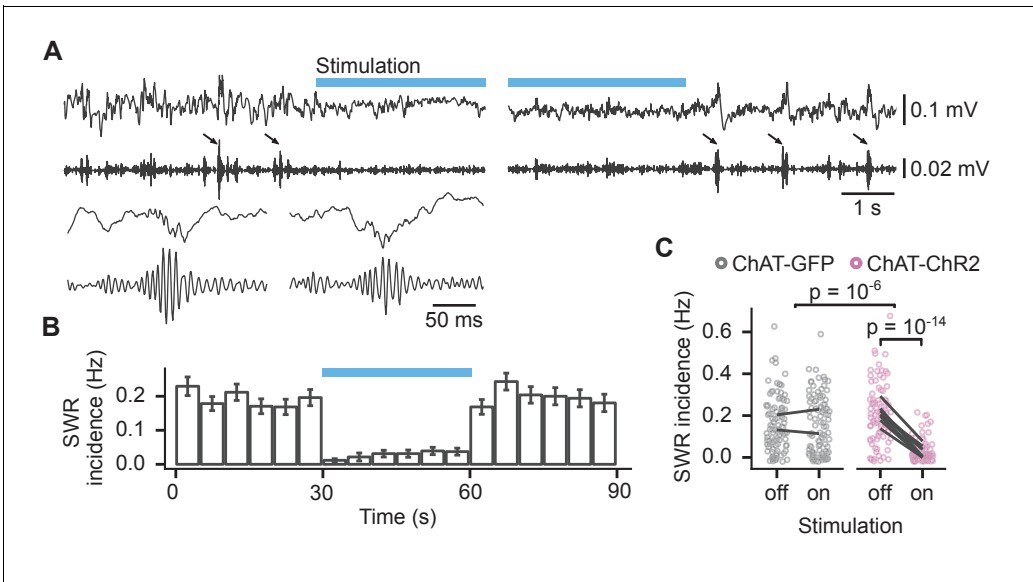

**Figure 5.** Activation of medial septum cholinergic neurons reduced incidence of sharp wave ripples (SWRs) during sleep. (A) Local field potential (LFP) from the CA1 of a sleeping mouse recorded before, during, and after optogenetic stimulation with 100–250 Hz bandpass filtered trace for ripple detection shown below. The detected SWRs are marked with arrows. The inset (lower left) shows two example SWRs at greater time resolution. (B) Histogram of SWR incidence before, during, and after 30 s of stimulation with 50-ms-long pulses at 10 Hz (n = 103 epochs from eight ChAT-ChR2 mice). (C) Comparison of SWR incidence during the stimulated and non-stimulated epochs for ChAT-GFP and ChAT-ChR2 mice. Lines connect mean incidence in individual mice. p-values were calculated with linear mixed-effects model for the interaction of mouse group–laser effects; groups were compared with post hoc test on least-square means.

The online version of this article includes the following source data and figure supplement(s) for figure 5:

**Source data 1.** Time of SWRs.
**Source data 2.** SWR incidence.
**Figure supplement 1.** Spectral peak frequency of ripples and duration of sharp wave ripples (SWRs) during sleep.
**Figure supplement 1—source data 1.** Ripple spectral peak frequency and duration.

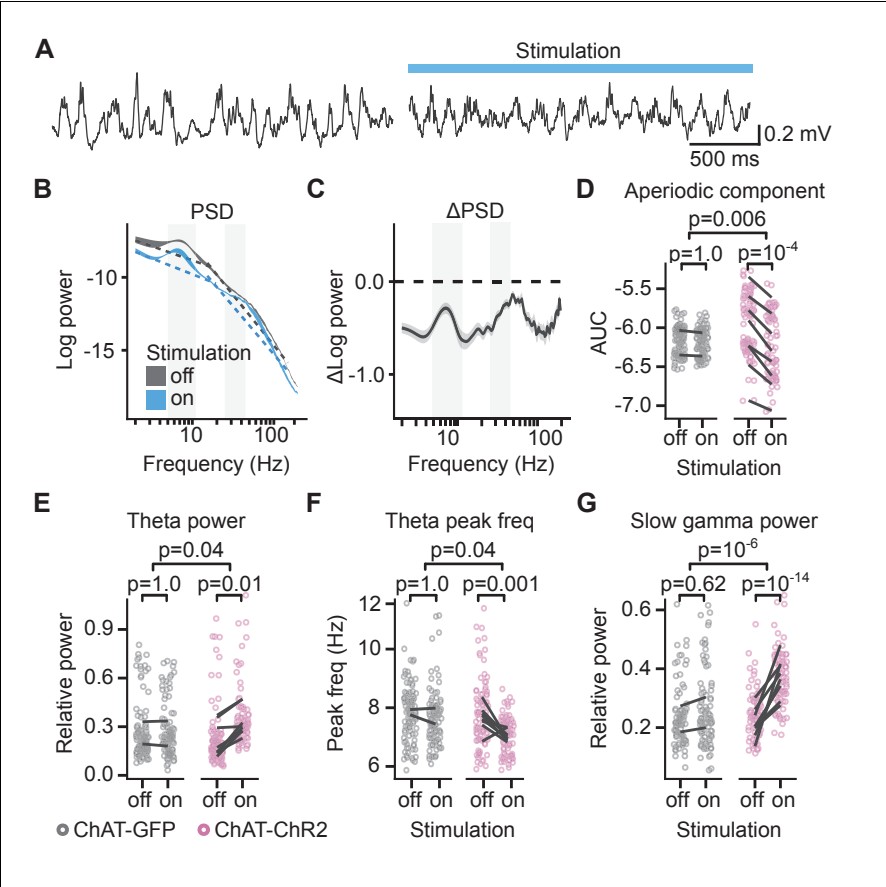

**Figure 6.** Cholinergic stimulation increased theta and slow gamma activity in sleeping mice. (**A**) Local field potential (LFP) recording from the CA1 of a sleeping mouse recorded without (left) and with (right) optogenetic stimulation. (**B**) Power spectral density (PSD) of the LFP recorded from a single animal during the epochs without and with optogenetic stimulation. Ribbons extend ±1 SEM of PSD. Gray background marks the frequency range of theta and slow gamma bands. The dashed lines show the fitted aperiodic component. (**C**) Difference in PSD between subsequent epochs with stimulation off and on. Data shown for the animal in (**B**). Ribbons extend ±1 SEM of log power. (**D**) Optogenetic stimulation in the choline acetyl transferase (ChAT)-ChR2 mice reduced the power of the aperiodic component in 15–150 Hz frequency range. (**E**) Optogenetic stimulation in the ChAT-ChR2 mice, but not in the ChAT-GFP mice, increased relative theta power, (**F**) decreased spectral peak frequency in the theta band, and (**G**) increased relative slow gamma power. Values were calculated on individual epochs, lines connect means for individual animals. p-values were calculated with linear mixed-effects model for the interaction of mouse group–laser effects; groups were compared with post hoc test on least square means.

The online version of this article includes the following source data and figure supplement(s) for figure 6:

**Source data 1.** AUC of aperiodic component power.
**Source data 2.** Theta power.
**Source data 3.** Theta peak frequency.
**Source data 4.** Slow gamma power.
**Figure supplement 1.** Power spectral density (PSD) of local field potential (LFP) in individual sleeping mice.
**Figure supplement 2.** Power spectral density (PSD) change in theta and gamma vs. neighboring frequency band.
**Figure supplement 2—source data 1.** PSD change per frequency band.

Broadband power of the aperiodic component decreased with light stimulation in the ChAT-ChR2 but not in the ChAT-GFP mice (linear mixed-effects model on area under curve [AUC] of estimated aperiodic component on PSD log-log plot, significant mouse group–laser interaction: $F_{(1, 4.6)}$=22, p=0.006, n = 369 epochs from 10 mice; AUC significantly decreased from $-5.96 \pm 0.04$ to $-6.23 \pm 0.04$, post hoc test for laser effect in ChAT-ChR2: $t_{(7.1)}$ = 10.1, p=$10^{-4}$, *Figure 6D*). Even though theta power was low during sleep outside of REM sleep, $99 \pm 0.1\%$ of the control and $100 \pm$

0% of the stimulated epochs had a relative theta peak. Optogenetic stimulation had a significantly different effect in the ChAT-GFP and the ChAT-ChR2 animals on the relative theta power (log-linear mixed-effects model, mouse group–laser interaction: $F_{(1, 4.8)}$=7.3, p=0.04, n = 368 epochs with theta peak from 10 animals, *Figure 6E*). In the ChAT-ChR2 mice, the power increased by $51 \pm 9\%$ (post hoc test: $t_{(9)}$ = 4.8, p=0.01) and the spectral peak frequency in the theta band decreased from $7.7 \pm 0.2$ to $7.2 \pm 0.1$ Hz (log-linear mixed-effects model, mouse group–laser interaction: $F_{(1, 4.8)}$=7.3, p=0.04, *Figure 6F*, post hoc test: $t_{(30)}$ = 4.5, p=0.001). To independently confirm that the stimulation increased relative theta power, we looked at the difference in the PSD between subsequent epochs with the stimulation off and on (*Figure 6C*, *Figure 6—figure supplement 1*). In the ChAT-ChR2 mice, the negative change in the theta band was significantly smaller than in the 12–15 Hz band (linear mixed-effects model: $F_{(1, 10)}$=21, p=0.001, *Figure 6—figure supplement 2A*). We conclude that the stimulation reduced the power in the theta frequency band significantly less than  in higher frequency bands.

The stimulation also increased by $30 \pm 4\%$ oscillations in the slow gamma band (25–45 Hz) (log-linear mixed-effects model: mouse group–laser interaction: $F_{(1, 232)}$=26, p=$10^{-6}$, n = 338 epochs with slow gamma peak from 10 animals, post hoc test for laser effect in ChAT-ChR2: $t_{(295)}$ = $-8.2$, p=$10^{-14}$, *Figure 6G*), while the spectral peak frequency in the slow gamma of $38 \pm 1$ Hz did not significantly change (linear mixed-effects model, mouse group–laser interaction: $F_{(1,10)}$ = 0.4, p=0.57). We independently confirmed the increase in relative slow gamma power by looking at the PSD change between subsequent epochs with the stimulation off and on. In the ChAT-ChR2 mice, the negative change of power in the slow gamma band was significantly smaller than in the 12–15 Hz band (mouse group effect in the linear mixed-effects model: compared to the 12–15 Hz band: $F_{(1, 8)}$=35, p=$10^{-4}$; compared to the 90–110 Hz band: $F_{(1, 10)}$=3.7, p=0.08, *Figure 6—figure supplement 2B*).

Our results demonstrate that in sleeping mice, optogenetic stimulation of MS cholinergic neurons promotes theta-gamma oscillations in the CA1, an effect that was not seen in awake mice. This suggests a difference between the effect of cholinergic stimulation in the sleeping and awake behaving animal.

## Discussion

Using optogenetics, we investigated the effects of stimulating MS cholinergic neurons on learning and hippocampal LFPs when delivered at different phases of an appetitively motivated spatial memory task. We found that: (1) MS cholinergic activation at the goal location, but not during navigation, impairs spatial memory formation; (2) MS cholinergic stimulation at the reward location reduces SWR incidence; and (3) cholinergic stimulation reduces SWR incidence and promotes theta-gamma rhythm in the sleeping mouse. These results show that timely control of cholinergic modulation is important for spatial learning on a time scale of seconds.

Our results indicate that cholinergic stimulation almost completely suppresses SWRs in sleeping animals and suppresses SWRs by about one half in awake, behaving animals. SWRs at the rewarded locations are thought to be crucial for learning (*Dupret et al., 2010*). Their suppression at the goal location in the experiments with the same stimulation protocol as that used in mice during learning suggests a possible explanation for the learning deficit induced by inappropriately timed cholinergic activity. Moreover, the effect of cholinergic stimulation on theta-gamma oscillations, which was prominent during sleep, was not observed when we applied the same stimulation at the goal location during learning, suggesting that learning was impaired through a mechanism independent of theta-gamma oscillations.

### Importance of timely regulation of cholinergic tone for memory formation

We found that temporally controlled optogenetic stimulation of MS cholinergic neurons could affect learning of the appetitive Y-maze task. Stimulation of cholinergic neurons during navigation did not affect the performance, while, strikingly, cholinergic stimulation in the goal zone significantly impaired task acquisition (*Figure 2*). The stimulation duration differed between the groups: it was longest in the 'throughout' group, followed by 'goal' and by 'navigation' group. The only significant impairment of task acquisition was seen in the 'goal' group, indicating that it was cholinergic

activation at the goal location that interfered with memory (*Figure 2C,E*). It may appear surprising that we did not also see a significant impairment with cholinergic stimulation throughout the task. However, the task performance in the 'throughout' group was not significantly different from the 'goal' group. We cannot exclude the possibility that prolonged optogenetic stimulation becomes less effective over time, either because the MS neurons become less activated or because vesicular ACh might be depleted with prolonged stimulation.

The lack of behavioral effect of the stimulation during the navigation phase, when the cholinergic tone is naturally high (*Fadda et al., 2000*; *Giovannini et al., 2001*; *Fadel, 2011*), may suggest that release of ACh in the hippocampus is already optimal or maximal, or that ACh receptors are saturated. MS cholinergic neurons are slow spiking neurons with a maximal rate of ~10 Hz during active exploration (*Ma et al., 2020*), the stimulation frequency used here. Thus, it is plausible that ACh receptor activation in the hippocampus had already reached a plateau, which our stimulation protocol would not increase further. The lack of behavioral effect of the stimulation during the navigation phase suggests that the effect of optogenetic stimulation was short-lived and restricted to the stimulation period, albeit with a short period of sustained activity following the stimulation (*Figure 1B*). This observation supports the idea that cholinergic modulation is timely controlled, but further experiments, for instance using ACh sensors in the hippocampus (*Jing et al., 2020*), will be necessary to confirm this hypothesis.

The impairment of memory formation by cholinergic stimulation in the goal zone, where the cholinergic tone is naturally lower (*Fadda et al., 2000*; *Giovannini et al., 2001*; *Fadel, 2011*), suggests that any potential beneficial effect of increased excitability or synaptic plasticity is outweighed by a requirement of reduced cholinergic activity. An interesting complementary experiment would be to silence cholinergic inputs during navigation or at the goal location to further explore the role of cholinergic tone during memory formation. There is evidence to suggest that CA1 SWRs, which occur during low cholinergic activity, play a crucial role in memory formation: disruption of SWRs in the first 15–60 min following training impairs learning of spatial navigation tasks (*Girardeau et al., 2009*; *Ego-Stengel and Wilson, 2010*), while their disruption or prolongation during the continuous alternation task impairs or improves learning respectively (*Jadhav et al., 2012*; *Fernández-Ruiz et al., 2019*). In exploring animals, SWRs occur during transient immobility periods, including periods at goal locations (*O'Neill et al., 2006*; *Dupret et al., 2010*; *Roux et al., 2017*). These SWRs stabilize spatial representations of the CA1 place cells supporting navigation toward the newly learned goals (*Roux et al., 2017*) and are predictive of performance in a spatial memory task (*O'Neill et al., 2006*; *Dupret et al., 2010*). During these SWRs, sequences of neuronal activation are replayed in both forward and reverse order (*Foster and Wilson, 2006*; *Csicsvari et al., 2007*; *Diba and Buzsáki, 2007*; *Karlsson and Frank, 2009*; *Ambrose et al., 2016*). We found that MS cholinergic activation for the brief time the mice spent in the reward zone, shorter than 50 s (95% percentile of stimulation duration), is sufficient to significantly impair memory formation in the Y-maze task (*Figure 2*). Therefore, we speculate that disruption of the normally occurring replay events in the reward zone is sufficient to impair long-term memory formation (*Figure 5*). However, selective disruption of SWRs at the reward zone did not affect rats' performance in the inbound phase of the W-maze task (*Jadhav et al., 2012*), which is comparable to the Y-maze task. In both of these tasks, animals could use either an allocentric place strategy or an egocentric rule-based strategy, or a combination thereof, and the relative importance of each could lead to differences in their reliance on SWRs. Alternatively, additional effects of MS cholinergic activation on intracellular signaling cascades and synaptic plasticity (*Brzosko et al., 2019*), synaptic inhibition (*Hasselmo and Sarter, 2011*; *Haam and Yakel, 2017*), or interference with extra-hippocampal reward-related signaling cannot be ruled out at this stage.

Because learning can be affected by the interruption of SWRs during post-learning sleep (*Girardeau et al., 2009*), and because our cholinergic activation during sleep achieves a similar effect on the SWRs (*Figure 3*; *Ma et al., 2020*), it would be of interest to see if the cholinergic activation during post-learning sleep would also impair spatial learning. This would show whether low cholinergic states are important also for memory consolidation during sleep and provide further evidence for a role of SWRs in memory.

## Cholinergic influence on hippocampal network activity

Hippocampal network activity varies with cholinergic tone and MS cholinergic neuron activity. MS cholinergic neurons discharge at a maximal rate when the animal is running (*Ma et al., 2020*), which corresponds to the highest theta power intensity in the CA1 and highest cholinergic tone measured in the pyramidal cell layer of CA1 (*Fadda et al., 2000*; *Fadel, 2011*). Conversely, cholinergic tone and MS cholinergic neuron discharge are at their lowest during slow-wave sleep and wake immobility, which are associated with the highest ripple incidence (*Fadda et al., 2000*; *Ma et al., 2020*). In accordance with these observations, we found that stimulation of MS cholinergic neurons reduces SWR incidence in both awake behaving animals and naturally sleeping animals, consistent with previous reports (*Figures 4* and *5*; *Vandecasteele et al., 2014*; *Zhou et al., 2019*; *Ma et al., 2020*).

We observed that stimulation of MS cholinergic neurons of sleeping mice causes an apparent decrease of the PSD across the entire frequency spectrum (*Figure 6*). A similar effect was reported previously for anesthetized and freely behaving animals (*Vandecasteele et al., 2014*). Signal decomposition into aperiodic and periodic components (*Donoghue et al., 2020*) showed that the optogenetic stimulation enhanced the periodic components with peaks in the theta and slow gamma bands and decreased the aperiodic component of the signal (1/f background). Our observation of the enhanced theta-gamma activity might appear at odds with previous reports that such manipulation does not change theta-gamma power during sleep (*Ma et al., 2020*) and quiet wakefulness (*Zhou et al., 2019*). The combined effect of the cholinergic stimulation on the periodic and aperiodic signal sums to near-zero values, which could explain the different conclusions, showing the advantage of PSD decomposition (*Donoghue et al., 2020*) when assessing the power of periodic signals.

Pharmacological evidence in vivo indicates that there are two distinct mechanisms of theta oscillations in the hippocampus, an atropine-sensitive and an atropine-resistant component (*Petsche et al., 1962*; *Buzsáki, 2002*; *Colgin, 2013*). The atropine-sensitive component is mediated by the combination of cholinergic and GABAergic neurons in the MS (*Buzsáki, 2002*; *Manseau et al., 2008*) and is slower than the atropine-insensitive theta rhythm, which is generated primarily by the entorhinal cortex (*Buzsáki, 2002*; *Colgin, 2013*). Moreover, atropine-sensitive theta was best detected in the anesthetized animal, while atropine-insensitive theta was shown to predominate in the running animal (*Kramis et al., 1975*; *Newman et al., 2013*). Consistent with this division, MS cholinergic stimulation in sleeping mice, in addition to increasing theta power, shifted the spectral peak in the theta band to a lower frequency (*Figure 6F*). Both effects were limited to the ChAT-ChR2 animals. The stimulation frequency of 10 Hz provided faster activation than the kinetics of metabotropic muscarinic receptors. Therefore, we did not expect to observe indirect effects on the network activity mirroring the stimulation frequency. Indeed, the spectral peak frequency in the theta band was lower than the stimulation frequency, and PSD did not show a spectral peak at 10 Hz (*Figure 6F*, *Figure 6—figure supplement 1*). In behaving mice, the MS cholinergic stimulation at the goal location did not have a significantly different effect on theta power and spectral peak in ChAT-GFP and ChAT-ChR2 mice (*Figure 3E,F*) and we observed in both groups of animals an increase of theta power. The lack of effect on theta-gamma rhythm during the memory task could be explained by the prominence of an atropine-resistant entorhinal-driven theta that would override any atropine-sensitive theta. It is also possible that the small sample of control animals (n = 2) has prevented us from detecting a subtle theta power change. Alternatively, a diminishing efficacy with the prolonged optogenetic stimulation could have prevented us from detecting a change in the theta-gamma oscillations. However, we did observe a reduction of SWR incidence at the goal location for the entire duration of the stimulation, suggesting that any decrease in the stimulation efficacy would be biologically minor.

## Possible implications for neurodegenerative disorders

Loss of cholinergic neurons in the basal forebrain is one of the hallmarks of AD (*Whitehouse et al., 1982*; *Bowen et al., 1982*), which is also associated with a reduction of ACh transporter expression in most cortical and subcortical areas (*Davies and Maloney, 1976*). These observations have led to the cholinergic hypothesis of AD, which suggests that loss of cholinergic inputs plays a role in the cognitive impairment of AD patients. However, the association between the loss of basal forebrain cholinergic neurons and AD is not completely clear (*Mesulam, 2004*), and a growing body of anatomical and functional studies suggests that MS cholinergic neuronal loss occurs in both healthy

aging and AD brain (*Schliebs and Arendt, 2011*; *Hampel et al., 2018*). Drugs compensating for the decline of cholinergic tone are seen as a rational treatment of aging-related memory loss and AD. However, so far, drugs targeting the cholinergic systems have shown limited beneficial effects on cognitive deficits of aging and AD, but the reasons why are not entirely clear (*Farlow et al., 2010*; *Ehret and Chamberlin, 2015*). Our results shed some light on one possible reason why cholinergic drugs have largely failed to improve the cognitive impairments in AD. Cholinesterase inhibitors prolong cholinergic activity by ~100 times and increase the basal cholinergic tone in the absence of spontaneous activity (*Hay et al., 2016*). In rodents, the cholinergic tone is high during exploration, which maintains the 'online' hippocampal state dominated by theta and gamma oscillations (*Buzsáki, 1989*; *Fadda et al., 2000*; *Giovannini et al., 2001*). Cholinesterase inhibitors may maintain a high cholinergic tone, preventing the network from transitioning into an SWR-dominant state. This could impair memory formation as our results suggest that artificially increasing ACh release for as short a time as 50 s during a low cholinergic state is sufficient to impair task acquisition (*Figure 2*). Moreover, enhancing cholinergic activity during the 'online' state did not bring beneficial effects to memory formation. Thus, our results suggest that suboptimal timing of cholinergic activity impairs long-term memory formation and supports the idea that appropriate timing of cholinergic modulation is crucial in learning and memory (*Micheau and Marighetto, 2011*).

## Materials and methods

### Key resources table

| Reagent type (species) or resource | Designation | Source or reference | Identifiers | Additional information |
|---|---|---|---|---|
| Genetic reagent (*Mus musculus*) | ChAT-Cre | The Jackson Laboratory | Cat. #: 006410; RRID:MGI:3689420 | Dr Bradford Lowell |
| Genetic reagent (*Mus musculus*) | Ai32 | The Jackson Laboratory | Cat. #: 012569; RRID:MGI:104735 | Hongkui Zeng, Allen Institute for Brain Science |
| Antibody | Anti-GFP (chicken polyclonal) | Abcam | Cat. #: AB13970; RRID:AB_300798 | IHC (1:1000) |
| Antibody | Anti-ChAT (goat polyclonal) | Millipore | Cat. #: AB144; RRID:AB_90650 | IHC (1:500) |
| Antibody | Anti-chicken Alexafluor488 | Life Technologies | Cat. #: A11039; RRID:AB_142924 | IHC (1:400) |
| Antibody | Anti-goat Alexafluor594 | Abcam | Cat. #: AB150132; RRID:AB_2810222 | IHC (1:1000) |
| Software, algorithm | R | R Project for Statistical Computing | RRID:SCR_001905 | |
| Software, algorithm | MATLAB | MATLAB | RRID:SCR_001622 | |
| Software, algorithm | IGOR Pro | IGOR Pro | RRID:SCR_000325 | |

### Animals

A total of 16 adult male WT (C57Bl/6), 51 adult male ChAT-Ai32 mice, and 5 *ChAT-Cre* mice were used in this study. ChAT-Ai32 mice were bred from *ChAT-Cre* mice that express Cre-recombinase under the control of the ChAT promoter (*ChAT-Cre*, Jackson Labs strain #006410; RRID:MGI:3689420) and mice of the Cre-reporter *Ai32* line (Jackson Labs strain #012569; RRID:MGI:104735), which carries a Cre-dependent, enhanced YFP-tagged channelrhodopsin-2 (ChR2-eYFP)-containing expression cassette (*Madisen et al., 2012*). All animal experiments were performed under the Animals (Scientific Procedures) Act 1986 Amendment Regulations 2012 following ethical review by the University of Cambridge Animal Welfare and Ethical Review Body (AWERB) under personal and project licenses held by the authors.

### In vivo electrophysiology in anesthetized mice

Mice were anesthetized with intraperitoneal injection of 1.2 g kg$^{-1}$ urethane and their head was secured in a stereotaxic frame. Body temperature was maintained at 35 ± 1°C with a heating pad. The head was shaved, and the skin opened; Bregma and Lambda were aligned horizontally, and craniotomies were made above the MS and CA3. Simultaneous optical activation in the MS (AP +1 mm, ML 0 mm, DV −3.6 mm, coordinates from Bregma) with a stripped optic fiber (200 µm, 0.22 NA; Doric Lenses) and electrical recordings in the MS or in the hippocampus (ML +2.4 mm, AP −2.46 mm, DV −2.5 mm) using an extracellular parylene-C insulated tungsten microelectrode (127 µm diameter, 1 MΩ; A-M Systems) were performed.

### Surgery

Surgeries were carried out following minimal standard for aseptic surgery. Meloxicam (2 mg kg$^{-1}$ intraperitoneal) was administered as analgesic 30 min prior to surgery initiation. Mice were anesthetized with isoflurane (5% induction, 1–2% maintenance, Abbott Ltd, Maidenhead, UK) mixed with oxygen as carrier gas (flow rate 1.0–2.0 L min$^{-1}$) and placed in a stereotaxic frame (David Kopf Instruments, Tujunga, CA). The skull was exposed after skin incision and Bregma and Lambda were aligned horizontally. A hole was drilled above the MS at coordinates AP +1 mm and ML 0 mm, and an optic fiber (200 µm, 0.22 NA; Doric Lenses) was lowered toward the MS (DV −3.6 mm) at low speed (1 mm min$^{-1}$). Once positioned just above the MS, the optic fiber was secured to the skull using dental cement (Prestige Dental).

Five ChAT-Cre mice underwent viral transduction of MS cholinergic neurons upon injection of viral particles. Two mice were injected with 0.5 µL of AAV5/9-EF1a-dio-EGFP-WPRE and three with 0.5 µL of AAV5/9-EF1a-dio-ChR2(H134R)-EYFP-WPRE (titers ranging 1.2–13.10$^{12}$ vg mL$^{-1}$; UNC Vector Core, Chapel Hill, NC), which were delivered through a metal cannula fixed to a 5 µL Hamilton syringe.

To perform recordings in freely moving animals, we implanted 10 mice with paired wire LFP electrodes, each consisting of two twisted 75 µm Teflon-coated silver wires (AGT0510, World Precision Instruments, Hitchin, UK) with tips spaced 150–300 µm and with one tip in the pyramidal cell layer. Mice were implanted bilaterally in CA1 (AP −1.7, ML ±1.2, DV 1 and 1.35, DV being taken from the surface of the brain). Ground and reference silver wires were connected to a stainless microscrew implanted over the cerebellum: AP −5.3, ML ±1.5. To record the electromyogram activity, a 75 µm Teflon-coated silver wire was implanted in the neck muscle. All wires were connected to a 32 pins Omnetics connector (Genalog, Cranbrook, UK). The exposed brain was covered with a protective dura gel (Cambridge NeuroTech, Cambridge, UK) to prevent damage upon cementing of the electrodes. LFP electrodes were individually glued to the skull using UV-cured glue (Tetric EvoFlow) and the implant was secured to the skull using dental cement (Super-Bond C and B; Prestige Dental, Bradford, UK). At the end of the implantation, 300–500 µL saline was injected subcutaneously for hydration and animals were placed in a post-surgery chamber at 34°C until full recovery from anesthesia. The mice were allowed to recover for 5 days before habituation started and during these 5 days were daily monitored and given oral Meloxicam as analgesic.

### Appetitive Y-maze task

Long-term spatial memory was assessed using the appetitive Y-maze task, as described in full by *Shipton et al., 2014*. Mice had to learn to find a food reward (condensed milk) on a three-arm maze that remained at a fixed location in relation to visual cues in the room. The three-arm maze, elevated 82 cm from the floor, consisted of gray-painted 50 × 13 cm arms bordered by 1 cm high white plastic walls, which extended from a central triangular platform. Plastic food wells (1.5 cm high) were positioned 5 cm from the distal end of the arms. Mice were kept on a restricted feeding schedule, allowing them to maintain at least 85% of their free food body weight. Before testing, the mice were habituated to the food reward and the maze in a different room to where behavioral testing would occur. During testing, mice were only allowed to make one arm-choice each trial and were only allowed to consume the reward if the correct arm was chosen, otherwise mice were removed from the maze and the trial was ended. Target arm assignments were counterbalanced such that at least one mouse of each experimental group was designated to each arm. Each mouse received 10 trials per day for 6–10 consecutive days, five starts from the left of the target arm and five starts from the

right in a pseudo-random order with no more than three consecutive starts from the left or right. The interval between the within-day trials averaged 10 min. The maze was rotated either clockwise or anticlockwise after each trial to discourage the use of intra-maze cues to help solve the task. Optogenetic stimulation started either from the beginning of the trial (navigation and throughout cohorts) or when the mouse reached the goal zone (goal cohort). Light stimulation ceased when the mouse reached the goal zone for the navigation cohort. Light stimulation was performed using a blue laser at 473 nm (Ciel, Laser Quantum, Cheshire, UK), powered at 25 ± 1 mW with 50-ms-long pulses at 10 Hz. Stimulation was controlled using custom-made procedures in Igor Pro (WaveMetrics, Lake Oswego, OR; RRID:SCR_000325).

## Optogenetic stimulation and electrophysiological recordings

Data were acquired from five ChAT-Ai32 male mice, three *ChAT-Cre* mice expressing ChR2 (both referred to as ChAT-ChR2) and two *ChAT-Cre* mice expressing GFP in the MS (ChAT-GFP). The mice were implanted with LFP electrodes for electrophysiological recordings and optic fiber for optogenetic stimulation. These mice were recorded during sleep and while performing the appetitive Y-maze task.

For recordings during sleep, after connecting the electrodes to the Whisper acquisition system (Neural Circuits, LLC, Ashburn, VA) and optic fiber to the laser, the animals were placed in a cage (different to their home cage), to which the animal was habituated over a period of 2 days. The floor of the cage was covered with standard bedding. The recordings started after the mice visibly stopped moving and consisted of 30-s-long laser stimulation at 473 nm, power 25 ± 1 mW using 50-ms-long pulses at 10 Hz alternating with 60–120 s interval without the stimulation. An overhead webcam camera tracked the movement and position of the animal. The videos were manually reviewed together with the recorded EMG signal to exclude trials that were interrupted by the mice moving.

For Y-maze task, the mice underwent the same habituation and learning protocol as described above in the appetitive Y-maze task section. During learning, mice were connected to the laser and to the Whisper acquisition system and placed at the starting arm of the maze. The laser was activated in the goal zone on alternating trials to allow within-subject comparison. Data from these five mice were not used in the behavioral analysis as the stimulation protocol (50% of the trials) was different from that used in behavior only (stimulation performed for all trials).

The position of the animal was tracked with an overhead webcam and automatically extracted using custom procedures in *MATLAB, 2019*. All recordings were performed using the Whisper acquisition system sampling at 25 kHz, laser stimulation was triggered using custom-made procedures in Igor Pro and synchronized with the electrophysiological and webcam recordings.

## Electrophysiology data analysis

Data analysis was performed in *MATLAB, 2019*; RRID:SCR_001622 and R version 3.4.4 (*R Development Core Team, 2018*; RRID:SCR_001905). To reduce contamination by volume conducted signal, we used staggered wire electrodes targeting the CA1 with one electrode in the pyramidal cell layer, and the differential signal was used to enhance signal differences between the hippocampal layers. To remove noise artifacts caused by wire movement and muscle contractions, changes in the consecutive samples of the EMG signal were detected. If the change exceeded a threshold set to two standard deviations, a 500-ms-long window of the signal centered on the noise timestamp was removed. We conducted the analysis on one of the bilaterally implanted CA1 LFP electrodes, selected based on the quality of signals for both theta oscillations and ripples.

For ripple detection, we adapted the method from *Vandecasteele et al., 2014*. The signal was down-sampled to 1.25 kHz and 100–250 Hz bandpass filtered with Type II Chebyshev phase-preserving filter (filter order = 4, stopband attenuation = 20 dB). Next, the filtered signal was squared, mean-subtracted, and smoothed by applying a moving average with 10-ms-long window. Ripples were detected when the squared signal crossed two standard deviations for 20–300 ms duration and its peak crossed seven standard deviations. Spectral peak frequency of a ripple was estimated as the frequency with maximum value in the PSD estimated with multitaper method on the signal from the ripple start to end. Ripple incidence was calculated as the number of detected ripples divided by the duration of the recording.

PSD was estimated using Welch's method (MATLAB built-in *pwelch* function with 0.5 s window and 0.25 s overlap) for frequencies spanning the range from 1 to 200 Hz. To visualize instantaneous changes in PSD during Y-maze trials, spectrogram was created with continuous wavelet transform using Morlet wavelets (MATLAB built-in *cwt* function with default parameters). Throughout the study, we defined theta band as 5–12 Hz and slow gamma as 25–45 Hz. The slow gamma frequency upper bound was chosen to exclude any line noise contamination at 50 Hz.

To estimate relative theta and slow gamma power, we used the FOOOF tool (*Donoghue et al., 2020*; https://github.com/fooof-tools/fooof). It models the estimated PSD as the sum of an aperiodic component and Gaussian peaks in narrowband frequencies. The aperiodic component was fitted on the PSD log-log plot with a straight line, which corresponds to a pink noise-like (1/f) background. To minimize the model error – the difference between the actual and the modeled PSD – the aperiodic component was estimated in two frequency ranges separately (3–15 and 15–150 Hz). Relative theta and slow gamma peaks and their spectral peak frequencies were taken from the Gaussian peaks fitted above the aperiodic component.

## Statistical analysis

Statistical analysis was performed in R version 3.4.4 (*R Development Core Team, 2018*). Data are reported as mean ± SEM unless otherwise stated. For significance testing, the normality of the data was assessed by Shapiro-Wilk test and by inspection of the quantile-quantile plot. If the normality criterion was satisfied ($p > 0.05$), a parametric test (one-way ANOVA or *t*-test) was used, otherwise a non-parametric test (one-way ANOVA on ranks or Wilcoxon test) was used, as described in the Results section. Following a significant one-way ANOVA on ranks, differences between groups were tested using a Dunn post hoc test with Holm-Bonferroni correction. To distinguish between the absence of effects and inconclusive results, we calculated BFs for the behavioral results (*Keysers et al., 2020*). Bayesian ANOVA was conducted using JASP software with default priors. BFs were calculated as the ratio between the likelihood of the data given the model with the effect of mouse group vs. the intercept only model. The post hoc pairwise comparisons were conducted using Bayes t-test in JASP with Cauchy priors without correcting for multiple comparisons.

Only mice with SWRs at the goal location were included in the analysis of the optogenetic stimulation effects during the Y-maze task (n = 5 ChAT-ChR2 and n = 2 ChAT-GFP mice). The statistical analysis of the effects in sleeping mice was performed using all CA1-implanted mice (n = 8 ChAT-ChR2 and n = 2 ChAT-GFP mice).

The effects of the optogenetic stimulation were assessed with linear mixed-effects models. This method allows for correlated samples from trials repeated in the same mouse and allows for an unbalanced number of samples between mice. Laser stimulation ($L \in$ {0 for inactive, one for active}) and effect of the mice group ($G \in$ {0 for ChAT-GFP, one for ChAT-ChR2}) were fixed effects in the models; the random variable representing the animal (a) was treated as a random effect in the intercept and slope estimation. The quantity Y in the trial i for animal a was modeled as:

$$Y_i = \beta_0 + R_{0a} + (\beta_1 + R_{1a})^* L + \beta_2^* G + \epsilon_i,$$

where

- $\beta_0, \beta_1, \beta_2$, are linear regression coefficients for the fixed effects.
- $R_{0a}, R_{1a}$ are random effects: normally distributed animal-specific corrections for linear regression coefficients with zero mean and maximum-likelihood standard deviation estimated by the model.
- $\epsilon_i$ is a random error with a normal distribution with zero mean and maximum-likelihood standard deviation estimated by the model.

The residual errors were checked for the assumptions of the linear models: mean of zero, no correlation with the predicted values and homoscedasticity. To satisfy these assumptions, in some models, a log-linear model of variable Y was created instead, otherwise as described above. The significant interactions were reported and the post hoc tests were performed on differences in least-square means of the paired groups. The tests used Satterthwaite estimation of degrees of freedom and adjusted p-values using Holm-Bonferroni correction.

The linear mixed-effects models were built in R with package 'lme4' and p-values for the fixed effects were obtained using Satterthwaite estimation of degrees of freedom implemented in the

'lmerTest' R package. Least square means were calculated and tested with 'lsmeansLT' function from the same package.

## Histological processing

Animals were terminally anesthetized by intra-peritoneal injection of pentobarbital (533 mg kg$^{-1}$) and then transcardially perfused with phosphate-buffered saline (PBS) followed by 4% paraformaldehyde. Brains were removed and post-fixed for 24–48 hr, then rinsed and subsequently cryoprotected overnight in 30% (w/v) sucrose dissolved in PBS. Coronal sections of 30–40 μm thickness of the MS and hippocampus were cut using a microtome (Spencer Lens Co., Buffalo, NY).

To verify the expression of ChR2 fused with the eYFP tag or ArchT fused with the eGFP tag and visualize the location of cholinergic neurons, sections were immunostained for eYFP/eGFP and ChAT. After rinsing in PBS, sections were incubated for 1 hr in a blocking solution comprising PBS with 0.3% (w/v) Triton X-100% and 5% (w/v) donkey serum (Abcam) containing 1% (w/v) bovine serum (Sigma). Sections were then incubated for $\geq$15 hr at 4°C with chicken anti-GFP (1:1000, Abcam AB13970; RRID:AB_300798) and goat anti-ChAT (1:500, Millipore AB144; RRID:AB_90650) antibodies. The sections were then washed, followed by 2 hr of incubation in blocking solution containing anti-chicken Alexafluor488 (1:400; Life Technologies A11039; RRID:AB_142924) and anti-goat Alexafluor594 (1:1000, Abcam AB150132; RRID:AB_2810222) at room temperature. Finally, the sections were rinsed and mounted in Fluoroshield with DAPI (Sigma).

To identify the placement of the electrodes (aided by DiI application on electrodes prior their insertion in the brain) and optic fiber tracks for each mouse, sections containing evidence of the implants were selected and mounted in Fluoroshield (Sigma).

Sections were examined with a Leica Microsystems SP8 confocal microscope using the 10× and 20× magnification objectives. The eYFP$^+$/GFP$^+$ and ChAT$^+$ cells were quantified manually using the ImageJ software. The location at which the implant appeared the deepest was determined and used to plot the implant location on the corresponding section in a mouse brain atlas (*Franklin and Paxinos, 2007*).

## Data and code availability

Code used for the analysis and to generate the figures can be accessed on the authors' GitHub site: https://github.com/przemyslawj/ach-effect-on-hpc (*Jarzebowski et al., 2021*; copy archived at swh:1:rev:3d4f5f8cecf7e6cc1b4bee7713bc582d5797674b).

# Acknowledgements

We thank Drs Julija Krupic and Mohamady El-Gaby for introducing us to recordings in freely moving animals. The authors gratefully acknowledge the Cambridge Advanced Imaging Centre for their support and assistance in this work.

# Additional information

### Funding

| Funder | Grant reference number | Author |
|---|---|---|
| Biotechnology and Biological Sciences Research Council | BB/N019008/1 | Ole Paulsen |
| Biotechnology and Biological Sciences Research Council | BB/P019560/1 | Ole Paulsen |

The funders had no role in study design, data collection and interpretation, or the decision to submit the work for publication.

### Author contributions

Przemyslaw Jarzebowski, Conceptualization, Data curation, Software, Formal analysis, Validation, Investigation, Visualization, Methodology, Writing - original draft, Project administration, Writing - review and editing; Clara S Tang, Conceptualization, Formal analysis, Investigation, Methodology;

Ole Paulsen, Conceptualization, Supervision, Funding acquisition, Writing - original draft, Project administration, Writing - review and editing; Y Audrey Hay, Conceptualization, Supervision, Investigation, Writing - original draft, Project administration, Writing - review and editing

### Author ORCIDs
Przemyslaw Jarzebowski (iD) https://orcid.org/0000-0001-6333-222X
Ole Paulsen (iD) http://orcid.org/0000-0002-2258-5455
Y Audrey Hay (iD) https://orcid.org/0000-0001-7765-5222

### Ethics
Animal experimentation: All animal experiments were performed under the Animals (Scientific Procedures) Act 1986 Amendment Regulations 2012 following ethical review by the University of Cambridge Animal Welfare and Ethical Review Body (AWERB) under personal and project licenses held by the authors.

### Decision letter and Author response
Decision letter https://doi.org/10.7554/eLife.65998.sa1
Author response https://doi.org/10.7554/eLife.65998.sa2

## Additional files

### Supplementary files
• Transparent reporting form

### Data availability
Code used for the analysis and to generate the figures can be accessed on the authors' GitHub site: https://github.com/przemyslawj/ach-effect-on-hpc (copy archived at https://archive.softwareheritage.org/swh:1:rev:3d4f5f8cecf7e6cc1b4bee7713bc582d5797674b/). Raw data are available on Zenodo.

The following dataset was generated:

| Author(s) | Year | Dataset title | Dataset URL | Database and Identifier |
|---|---|---|---|---|
| Jarzebowski P, Tang CS, Paulsen O, Hay YA | 2021 | Impaired spatial learning and suppression of sharp wave ripples by cholinergic activation at the goal location | https://zenodo.org/record/4708331#.YIKw7qlKhpQ | Zenodo, 4708331#.YIKw7qlKhpQ |

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
