## [Decision Letter]

**Acceptance summary:**

This paper is of interest for those interested in the roles of cholinergic projections from the medial septum and sharp wave-ripples on reward learning. The work provides compelling evidence showing that activation of septal cholinergic cells at reward zones suppresses sharp wave-ripples and impairs behavioral performance in freely behaving animals. The work extends our knowledge of the effect of medial septum cholinergic projections on spatial memory.

**Decision letter after peer review:**

[Editors’ note: the authors submitted for reconsideration following the decision after peer review. What follows is the decision letter after the first round of review.]

Thank you for submitting your work entitled "Cholinergic suppression of sharp wave-ripples impairs hippocampus-dependent spatial memory" for consideration by *eLife*. Your article has been reviewed by 3 peer reviewers, one of whom is a member of our Board of Reviewing Editors, and the evaluation has been overseen by a Senior Editor. The following individuals involved in review of your submission have agreed to reveal their identity: Stefan Remy (Reviewer #2); Jiamin Xu (Reviewer #3).

Our decision has been reached after consultation between the reviewers. Based on these discussions and the individual reviews below, we regret to inform you that your work will not be considered further for publication in *eLife*.

Reviewers were enthusiastic about the question and the system that was studied. However, major concerns were raised about the paper in its current form. Reviewers were not convinced that the conclusions of the paper are robustly supported by the results. Specifically, reviewers raised concerns about the low sample size and other methodological aspects of the study. Reviewers felt that, for the results to fully support the conclusions, additional experimentation and additional control experiments would be required. Concerns were also raised about the novelty of some of the results. Some of the results provide insights that are similar to those reported in previous studies (e.g., Vandecasteele et al., 2014 and Ma et al., 2020), specifically regarding cholinergic suppression of hippocampal ripple oscillations and the involvement of muscarinic receptors in this effect.

We would be willing to consider a majorly revised version of this study in the future, should you choose to go that route.

Reviewer #1:

This paper attempts to address the interesting question of how activating septal cholinergic neurons affects learning and memory. Unfortunately, however, the conclusions are not strongly supported by the results.

1. Proper controls for the ontogenetic experiments are missing. For example, a light only control is only included for the goal vs. no stimulation comparison in the first part of the paper. No control for light artifacts is included in the last part of the paper in which theta and gamma are examined. 10 Hz stimulation is. used, yet there is no discussion of effects of light artifacts on spectral measurements within this range.

2. A major problem is that the duration of the light stimulation protocol differs across groups. The authors did not consider the possibility that effects were due to the circuitry being perturbed differently by different durations of stimulation (longer cholinergic stimulation vs. shorter cholinergic stimulation).

3. It is difficult to interpret the data from the "stimulation throughout" group. The authors did not statistically compare "goal stimulation" and "stimulation throughout" groups. If the interpretation is correct that activating MS cholinergic neurons around the goal impairs learning, then one would expect "stimulation throughout" to also impair learning since it also includes stimulation at the goal.

4. The authors do not provide convincing causal evidence that suppression of sharp waves around the goal causes impaired learning, although they seem to imply this throughout the paper.

5. Some of the main results in the paper have been reported previously. The SWR suppression due to cholinergic stimulation is not novel, as the authors acknowledge. Vandecasteele et al. (2014) showed this effect in behaving and urethane anesthetized animals, and Ma et al. (2020) showed it in sleep. Ma et al. also reported muscarinic receptor involvement in SWR suppression.

6. Are these methods able to reliably detect sharp wave ripples? Some muscle artifacts can look like sharp wave ripples. In the absence of depth profiles from a linear probe or detection of populations bursts from single units, it is difficult to evaluate the extent to which the authors' sharp wave-ripple detection methods are reliable.

Reviewer #2:

In this manuscript Jarzebowski et al. investigate the dynamical aspect of cholinergic modulation. The authors report phase-specific effect of optogenetic cholinergic modulation in the appetitive Y-maze long-term memory task, as well as the switches of the CA3 and CA1 activity from ripple activity to theta/gamma oscillations. This work builds on the previously published studies (Vandecasteele et al., 2014; Ma et al., 2020) that have already reported reduction in ripple incidence upon MS cholinergic stimulation. The novelty of this study lies in reporting the differences on learning dependent on the time (and location) of the simulation and apparent differences on the CA1 and CA3 ripples incidence. However, the conclusions in some cases are not strongly enough supported by the data. In my view several points need clarification and in particular the statistical underpowering resulting from very low number of animals in the results presented in Figure 5 speak against publication of the manuscript in the current form.

1. The results showing decrease in learning following the MS cholinergic stimulation (2C-E) in the goal zones are convincing. However, it is known that SWRs may differ substantially in length, frequency and amplitude dependent on the behavioural state of the animal (ex. Joo and Frank, Nat Neuro Review 2019), which may have specific relevance for memory. Thus, by stimulating both the rewarded and the non-rewarded goal zones the effects of the stimulation on SWR may have differential effects on SWRs in rewarded and non-rewarded locations. The authors should compare SWRs and the effects of stimulation in rewarded and non-rewarded goal zones. A question is whether effects on SWRs in the non-rewarded goal zone are similar or substantially different, in which case the interpretation of how stimulation may have mechanistically contributed to memory formation could be fundamentally different.

2. The authors show the reduction of ripples incidence in CA1 and CA3 upon MS cholinergic stimulation in both in sleeping and anesthetized animals. However, they also claim that "…MS cholinergic stimulation enhanced a scopolamine-sensitive theta oscillation in both anesthetized and sleeping mice (supplementary Figure 3)". To substantiate this claim by data, they need to block muscarinic receptors in sleeping animals to assure that it holds in both conditions. I did not find these data in the manuscript. Scopolamine sensitivity is a classical criterion for differentiation of types of theta oscillations so that this information would be highly relevant for the interpretation of the results with respect to effects of stimulation on theta oscillations. If the authors did not perform these experiments, they should not extend the interpretation to the sleep condition.

3. Figure 4: Using the method presented in Haller et al., 2018, the authors show an increase in theta power and gamma power in both CA1 and CA3, following MS cholinergic stimulation. While these results are interesting, it would be much more convincing if the authors showed the grouped data of all animals, including the error bars in the Figure 4B, instead of only the data obtained from an individual animal. More importantly the statistical analysis depends on a linear fit of the background spectrum intensity, which from inspecting the data seems moderately imprecise. Since the differences reported are rather small, the authors should use a 1/f polynomial fit (as also used in Haller et al., 2018) of the background spectrum intensity which would confirm the robustness of the described differences.

4. Related to Figure 4: The authors state: "The increase in the power was associated with significantly lower frequency of the theta peak in the CA1, but not in the CA3." It would be helpful for the reader to also show these results in the Figure.

5. The authors are reporting: "Slow gamma (25-45 Hz) increased in CA1 by 54 {plus minus} 6 % and in CA3 by 8 {plus minus} 8 % ". It has been previously shown that slow gamma is driven by CA3 and propagates to CA1 (Colgin et al., 2014, 2016, Schomburg et al., 2014). To my understanding, this would imply that the low gamma activity should reach CA1 with a smaller amplitude than measured in CA3. As the opposite is observed here, does that imply that the local CA1 network activity lead to amplification of the gamma power in CA1? This should be discussed.

6. In the Figure 5, the authors report decreases in CA1, but not in CA3 ripples incidence in the goal zone following the MS cholinergic stimulation. I have several difficulties with the interpretation of the results presented in this Figure. First, the authors state that only successful trials were analysed. As mentioned above already, given that the stimulation was performed also during the unsuccessful trials, the results obtained in the unrewarded goal zones should be presented. Furthermore, it is stated: "…n=3 mice included in the analysis with the ripple incidence at goal {greater than or equal to} 0.03". I think that "{less than or equal to} 0.03" is meant here. In any case, looking carefully at the Figure 5D, a decrease of the ripple incidence at goal can be observed in 2 of the 3 animals while the increase observed in one single animal accounts for no difference which underlies the conclusion that the effects are specific for CA1. This experiment is strongly underpowered and the conclusions are not convincingly supported by the data.

7. Furthermore, no difference in theta and gamma power in CA1, but increase in slow gamma in CA3 (5E-5G) are reported. Looking carefully at 5I, a trend in increase could be also observed in CA1. Also here, the number of animals must be increased.

8. The title of the paper is: "Cholinergic suppression of sharp wave-ripples impairs hippocampus dependent spatial memory" The authors are also showing the effect of MS cholinergic stimulation on the SWRs during sleep, but the link between this effect and spatial memory is not explored. It is known that supressing SWRs specifically during the post-learning sleep impairs spatial memory (Girardeau et al., Nature Neuroscience 2009). Thus, it would be interesting to see whether the MS cholinergic stimulation and SWRs suppression during sleep impairs spatial memory. This should be at least discussed.

Reviewer #3:

The study focused on the effect of MS cholinergic activity on hippocampal oscillations and extended our knowledge of cholinergic function in hippocampus-dependent spatial learning. Impairment of hippocampal ripple oscillations during awake immobility leads to significant performance deficit in the Y-maze task, highlighting the importance of precise timing of cholinergic input in memory formation.

1. The conclusion points 1/2/3 in the abstract maybe more coherent if rearranged to 3/1/2. But this might lead to structural re-organization of the article and the figures (suggested figure order: Figure 1-3-4-2-5). The logic behind this order is:

a. Optogenetic stimulation of MS cholinergic neurons impair hippocampal ripples during SWS (Figure 3/4).

b. What about ripples during awake immobility?

c. Optogenetic stimulation during goal period of the Y-maze impairs learning (Figure 2).

d. Such impairment was due to reduced CA1 ripple incidence (Figure 5).

2. In the introduction: "activation of MS cholinergic neurons switches the CA3 and CA1 network states from ripple activity to theta/gamma oscillations". The phrasing might be questionable

3. Page 18, first paragraph, the description seems a little bit redundant.

4. Figure 3A, it seems that theta oscillations emerge with the optogenetic stimulation, was the stimulation strictly delivered during SWS or was this particular theta represents REM states? It would be better if the authors also show the LFP and ripple trace after the light stimulation.

5. Page 22, last sentence of the first paragraph, "…while the peak frequency did not significantly change from 39 {plus minus} 1 Hz in the CA1 and 38 {plus minus} 2 Hz in the CA3…", is a little bit confusing, needs further explanation.

6. Page 22, second paragraph, relative theta and gamma power change was reported in CA3, how about CA1?

7. In previous reports (Vandecasteele, 2014 and Ma, 2020), MS cholinergic stimulation can completely inhibit hippocampal ripple oscillations. But in this study, there are still a lot of ripples not suppressed. Please explain the difference.

[Editors’ note: further revisions were suggested prior to acceptance, as described below.]

Thank you for submitting your article "Impaired spatial learning and suppression of sharp wave ripples by cholinergic activation at the goal location" for consideration by *eLife*. Your article has been reviewed by 3 peer reviewers, one of whom is a member of our Board of Reviewing Editors, and the evaluation has been overseen by Laura Colgin as the Senior Editor. The following individuals involved in review of your submission have agreed to reveal their identity: Jiamin Xu (Reviewer #2); Fabian Kloosterman (Reviewer #3).

Essential Revisions:

Some conclusions that are strongly stated in the current version will need supporting data in the future or will need to be toned down with caveats discussed. Please refer to the individual reviews below for specific details.

Reviewer #1 (Recommendations for the authors):

The authors satisfactorily addressed my prior major concerns. I have no major concerns remaining. I only have a few concerns remaining that can be easily addressed by the authors without affecting the major conclusions of the paper.

Page 5: "In hippocampal CA3, cholinergic activation induces a slow gamma rhythm primarily by activating M1 muscarinic receptors": It should be noted that these are in vitro studies. As this sentence is written currently, the comparisons between CA3 and CA1 are potentially misleading.

Page 10: "We did not detect an effect of the laser on the duration the mice spent at the goal location (linear mixed-effects model, mouse group – laser interaction: F(1, 78) = 0.01, p = 0.94, laser effect: F(1, 78) = 0.1, p = 0.73)." I am confused by this passage. If optogenetic stimulation at the reward location affects memory, as the authors claim, one would expect a differential effect of laser stimulation on the ChR2 mice compared to the GFP-only control mice (and a significant effect of the laser in the ChR2 group). Yet, a non-significant interaction effect is reported.

Page 13: "Because we detected few SWRs in the unrewarded trials, we restricted the statistical analysis of the effects of optogenetic stimulation to rewarded trials. The SWR incidence in the non-stimulated trials was not significantly different between early (before day 5) and late learning (linear mixed-effects model, effect of early vs late learning: F(1, 110) = 0.3, p = 0.58, Figure 4D).": The authors state that they restricted analysis of effects of optogenetic stimulation to rewarded trials but then analyze SWR incidence in the very next sentence and paragraph. Is there a typo here?

Reviewer #2 (Recommendations for the authors):

This submitted work is clearly better structured and well-controlled version of the previously submitted manuscript. The authors addressed most of the comments. I only have two concerns.

1. The optogenetic stimulation induced ripple inhibition effect at the goal location and during sleep seems a little bit inconsistent (Figure 4E and Figure 5C). The authors stated in the text the ripple inhibition statistics under the two conditions: ripple incidence was reduced from 0.11 {plus minus} 0.01 Hz to 0.05 {plus minus} 0.01 Hz when stimulated at the goal location, and from 0.21 {plus minus} 0.01 Hz to 0.03 {plus minus} 0.01 Hz during sleep. Also, from the authors' response to the review comments (Reviewer #3, major comment 7), "We report that SWR incidence in 4 ChAT-ChR2 decreased by 100% and for another 4 reduced by a median of 88 {plus minus} 2%.". I assume that the authors were referring to the situation during sleep (based on the statistics and the number of animals used in each condition), which means that the ripple inhibition effect would be somewhere around 50% during stimulation at the goal location. Please elaborate on this.

2. Although the analyses of the effect of optogenetic stimulation on hippocampal theta and gamma oscillations during sleep reveal some very interesting results, it seems that the relevance of these analyses with the key claim of the submitted work (as indicated by the title) is a little bit farfetched.

Reviewer #3 (Recommendations for the authors):

1. Figure 3B: the spectrogram shows high relative (z-scored) power for the high frequencies in start and center, but not goal. I would have expected to see at least some high power in the ripple band in the goal. Could the authors clarify how exactly the z-scoring was performed? If the spectrogram is an average across multiple trials, then this will tend to obscure transient, non-time locked oscillations like SWRs.

2. What is the effect of optical stimulation of cholinergic neurons on theta/gamma/SWRs in the "throughout" and "navigation" conditions? Are these effects consistent with the hypothesis that the learning deficit is caused by a reduction of SWRs at the goal location? Could additional insights be obtained into possible changes induced by stimulation (e.g. theta oscillations during navigation) that do *not* correlate with the learning deficit?

3. Page 19: the authors cite Jadhav et al. 2012 when stating "disruption of SWRs in the first 15 to 60 minutes following training impairs learning of spatial navigation tasks". However, Jadhav et al. disrupted SWRs during the training and not following the training.

4. Page 20: Both reverse and forward replay are observed during brief pauses or reward consumption in the awake state when animals explore a maze or learn a task. So, it is likely that in the reward zone in the Y-maze task one will observe both forward and reverse replay. While it is fine to speculate that disruption of reverse replay mediates the behavioral deficit, it cannot be based on the assumption that replay at the goal location is only of the reverse kind.

5. What is the time in between individual trials?

6. To characterize the learning in the Y-maze, the authors determine the day at which criterion is reached. This metric is rather coarse. Instead, the authors could fit a learning curve (e.g. sigmoid function) to the trial responses and estimate the learning rate for each animal. Furthermore, it would be informative to show individual learning curves for all animals, in addition to the average learning curves that are shown now.

7. To assess the effect of stimulation at the goal on hippocampal activity, the authors look at average SWR rate and average theta/gamma power. However, when the animals are in the goal region, they likely show a mixture of behavioral states that is associated with periods of theta and non-theta (incl. SWRs). Is more (or less) time spent in theta state during stimulation? Could it be that time spent in non-theta states is lower, but SWR rate within this state has not changed? Judging from the example in figure 4B, it may be the case that with stimulation the first SWR after arriving at the goal is delayed compared to no stimulation condition – is this consistent across all subjects?

---

## [Author Response]

[Editors’ note: the authors resubmitted a revised version of the paper for consideration. What follows is the authors’ response to the first round of review.]

Reviewer #1:This paper attempts to address the interesting question of how activating septal cholinergic neurons affects learning and memory. Unfortunately, however, the conclusions are not strongly supported by the results.1. Proper controls for the ontogenetic experiments are missing. For example, a light only control is only included for the goal vs. no stimulation comparison in the first part of the paper. No control for light artifacts is included in the last part of the paper in which theta and gamma are examined. 10 Hz stimulation is. used, yet there is no discussion of effects of light artifacts on spectral measurements within this range.

As pointed out by the Reviewer, the manuscript’s initial version did not control for potential artifacts of the optogenetic stimulation on theta-gamma activity or SWR events. To address this concern, we performed additional recordings during natural sleep and the Y-maze task from two ChAT-Cre mice injected with a GFP-expressing AAV in the MS. The results have been added to Figures 3-6 and the associated supplementary figures. To compare the effects of stimulation between the ChAT-GFP and ChAT-ChR2 mice, we have extended our linear mixed-effects model to include the effects of the stimulation, the mouse group, and their interaction. We observed that the effect of optogenetic stimulation on SWR incidence was different between the two mouse groups in the natural sleep and the memory task (significant interaction between the stimulation condition and the mouse group effects). The effect on the theta-gamma activity was different between the mouse groups in the natural sleep but not the memory task. We used Dunn post hoc test to estimate p-values for the effect of the stimulation within the groups. We amended the Methods and the Results sections for Figures 3-6 accordingly, and we rule out the potential problem that the stimulation frequency is within the theta frequency range in the Discussion section. We do not report all the changes here as the modifications have been extensive.

2. A major problem is that the duration of the light stimulation protocol differs across groups. The authors did not consider the possibility that effects were due to the circuitry being perturbed differently by different durations of stimulation (longer cholinergic stimulation vs. shorter cholinergic stimulation).

Indeed, the duration of the light stimulation differed between the mouse groups with the ‘navigation’ group receiving the shortest stimulation (8 ± 1 s), followed by ‘goal’ (34 ± 1 s) and by the ‘throughout’ stimulation (42 ± 1 s). The experimental design does not allow us to rule out a role of duration fully; however, if the duration was the decisive variable then we would expect light throughout the maze to induce more impairment than light only at goal location, which is the opposite of what we observed. This argument supports our claim that stimulation at different stages of the task has a differential effect. These results are now better described (page 8), and we added a paragraph to the Discussion about this potential problem (page 17-18).

Page 8: “Whilst the stimulation for the ‘goal’ group lasted longer than the ‘navigation’ group (34 ± 1 s vs 8 ± 1 s), the duration alone cannot explain the different effects of the optogenetic stimulation. The ‘throughout’ group received the longest stimulation (42 ± 1 s) but presented an intermediate learning curve: it was not significantly different from either the ‘no stimulation’ group (p = 0.28) or the ‘goal’ group (p = 0.54). Therefore, the spatial location in the maze where the optogenetic stimulation took place was most likely the factor that decided the behavioral outcome.”

and page 17-18: “The stimulation duration differed between the groups: it was longest in the ‘throughout’ group, followed by ‘goal’ and by ‘navigation’ group. The only significant impairment of task acquisition was seen in the ‘goal’ group, indicating that it was cholinergic activation at the goal location that interfered with memory (Figure 2C,E). It may appear surprising that we did not also see a significant impairment with cholinergic stimulation throughout the task. However, the task performance in the ‘throughout’ group was not significantly different from the ‘goal’ group. Nevertheless, we cannot exclude the possibility that prolonged optogenetic stimulation becomes less effective over time, either because the MS neurons become less activated or because vesicular ACh might be depleted with prolonged stimulation.”

3. It is difficult to interpret the data from the "stimulation throughout" group. The authors did not statistically compare "goal stimulation" and "stimulation throughout" groups. If the interpretation is correct that activating MS cholinergic neurons around the goal impairs learning, then one would expect "stimulation throughout" to also impair learning since it also includes stimulation at the goal.

We have now added the comparison between the ‘goal stimulation’ and ‘stimulation throughout’ groups to the main text, and we observed no statistical difference between the groups. This statistical result is consistent with the Reviewer’s interpretation that stimulation ‘throughout’ also impairs the learning of the task, and, indeed, although not statistically significant, we observed a trend towards delayed learning with stimulation throughout. One possible explanation for the apparently reduced effect of the stimulation in the “throughout” group could be that prolonged optogenetic stimulation becomes less effective over time, resulting in a reduced amount of ACh released when the mouse reaches the goal area.

We have now added the statistical comparison in the main text, which reads (page 8):

“The ‘throughout’ group received the longest stimulation (42 ± 1 s) but presented an intermediate learning curve: it was not significantly different from either the ‘no stimulation’ group (p = 0.28) or the ‘goal’ group (p = 0.54).”

We also discuss possible reasons why the ‘throughout’ group fails to show a significant difference from the ‘no stimulation’ group (page 18):

“It may appear surprising that we did not also see a significant impairment with cholinergic stimulation throughout the task. However, the task performance in the ‘throughout’ group was not significantly different from the ‘goal’ group. Nevertheless, we cannot exclude the possibility that prolonged optogenetic stimulation becomes less effective over time, either because the MS neurons become less activated or because vesicular ACh might be depleted with prolonged stimulation.”

4. The authors do not provide convincing causal evidence that suppression of sharp waves around the goal causes impaired learning, although they seem to imply this throughout the paper.

We agree with the Reviewer that our study provides correlational rather than causal evidence. We updated the manuscript, including the title, to remove any implication of a causal effect. Based on previous research on the SWRs role in memory and because SWR incidence reduction was the most prominent effect induced by ACh neuron stimulation on the CA1 LFP in the behaving mouse, we believe the most parsimonious explanation for the effect of ACh on spatial learning is through the reduced incidence of SWRs. However, and as discussed in the original manuscript, we now also discuss that ACh promotes synaptic depression (Brzosko et al., 2019) and synaptic inhibition (Hasselmo and Sarter, 2011; Haam and Yakel, 2017) and we cannot rule out here that ACh could act not only through changes in SWRs but also on plasticity or inhibition.

We changed the title of the article to reflect the findings better:

“Impaired spatial learning and suppression of sharp wave ripples by cholinergic activation at the goal location”

and changed the text on page 14, which now reads:

“Hence this result suggests that the reduced SWR incidence is a mechanism relevant for the memory impairment induced by cholinergic stimulation in this task.”

5. Some of the main results in the paper have been reported previously. The SWR suppression due to cholinergic stimulation is not novel, as the authors acknowledge. Vandecasteele et al. (2014) showed this effect in behaving and urethane anesthetized animals, and Ma et al. (2020) showed it in sleep. Ma et al. also reported muscarinic receptor involvement in SWR suppression.

It is correct that some of our results have been reported previously, which we acknowledge throughout the manuscript. Because the focus of our paper is on the learning impairment induced by cholinergic stimulation and the observation that this impairment correlates with a decrease of SWR incidence at goal location, we have now removed from the manuscript the replication of results for urethane anesthetized animals and removed the description of results on the role of muscarinic receptors. However, we kept results for naturally sleeping mice for two main reasons.

First, the results obtained in naturally sleeping animals are a useful comparison to the results obtained in the memory task, especially since a change in theta-gamma activity observed in the sleeping animals with cholinergic stimulation might also have been expected in the memory task.

Second, we believe that the use of an alternative analysis technique, one that distinguishes between periodic and aperiodic components of the PSD (originally referred to as Haller et al. bioRxiv 2018, now Donoghue et al., Nat Neurosci 2020), will be of significant interest to the readership since it may help resolve some apparent discrepancies in the field. Vandecasteele et al. (2014) report that MS stimulation ‘increased theta power in anesthetized mice but it decreased or had no effect on theta power in behaving mice’ and Ma et al. (2020) report in sleeping mice that ‘During the stimulation period, there was no obvious evidence of hippocampal theta oscillations … But we observed a significant change on hippocampal theta power as it *decreased* from 0.59 ± 0.17 to 0.41 ± 0.12’ (our italics). Distinguishing between periodic and aperiodic components of the PSD could help explain these non-intuitive results, which are only alluded to in Vandecasteele et al. and Ma et al. We observed a reduction of the aperiodic component of the PSD but an increase in the periodic component in the theta band by cholinergic stimulation in naturally sleeping animals (Figure 6). Thus, a reduction in the aperiodic component may cancel out the increase in the periodic component. Arguably, it is the periodic component that better reflects theta oscillatory activity. A similar argument could be made for the slow gamma oscillation.

We believe it is important to report these results, that they will be of interest, and that they deserve to be published.

6. Are these methods able to reliably detect sharp wave ripples? Some muscle artifacts can look like sharp wave ripples. In the absence of depth profiles from a linear probe or detection of populations bursts from single units, it is difficult to evaluate the extent to which the authors' sharp wave-ripple detection methods are reliable.

To record sharp wave ripples we used the signal from paired wire electrodes staggered in the dorso-ventral direction. The electrode placement allowed us to record ripple activity with phase-reversal, but we observed the sharp-waves only on some of the paired electrodes. As pointed out by the Reviewer, electrical noise or muscle movement can cause ripple-like profile in the signal. Therefore, we applied two methods that aim to limit the events we might otherwise incorrectly classify:

1. We coupled the LFP and the EMG recordings. Whenever we detected a high amplitude change in the EMG signal, we excluded any ripple-like events from the surrounding time window.

2. We subtracted the signal between paired wire electrodes whose tips were spaced 150-300 μm in the dorso-ventral direction. This procedure canceled out synchronous changes on both electrodes like those caused by electrical noise or muscle artifacts, and it strengthened the same frequency, the out-of-phase signal on both electrodes, e.g. due to ripple phase reversal.

Both of these methods ensure the detected events are phenomena in the CA1 local field potential.

The short bursts of fast oscillations could be sharp-wave-associated ripples or, as differentiated by some studies, fast gamma ripples (Sullivan et al. 2011). These two were described as having different physiological mechanisms and different spectral peak frequency (Sullivan et al. 2011). The spectral peak frequency for the detected ripples showed bimodality (Figure 5—figure supplement 1A) as previously described in Sullivan et al. 2011, and we changed our methods to only classify as SWRs the events with spectral peak frequency ≥140 Hz. The stimulation significantly decreased ripple incidence in sleep and the Y-maze task when tested for all ripples and tested for ripples with ≥140 spectral peak frequency.

Finally, we would like to comment that because the SWR detection methods rely on a set threshold for ripple detection, the distinction between ripple and non-ripple events is always somewhat arbitrary (Buzsaki et al. 2015).

We added a Figure 3—figure supplement 1 with examples demonstrating the signal from the paired electrodes, its processing, and the detected ripples. Figure 3—figure supplement 2A and Figure 5—supplement 1A show peak power frequency of all detected ripple events.

On page 10 we added the following about the LFP signal:

“We used staggered wire electrodes to record the field potentials and subtracted the signal in one electrode from that in the other. This subtraction procedure cancels out synchronous changes on both electrodes, like those caused by movement artifacts, and enhances locally generated phase-reversed signals, such as theta, gamma and ripple events.”

On pages 12-13 we added the following about the SWR detection:

“To identify the SWRs, we detected ripple events in the LFP and excluded any candidate ripples that co-occurred with electromyography (EMG)-detected muscle activity. Only ripples with spectral peak frequency ≥140 Hz were identified as SWRs (Sullivan et al. 2011, Figure 4A, Figure 4—figure supplement 1 and 2).”

Reviewer #2:In this manuscript Jarzebowski et al. investigate the dynamical aspect of cholinergic modulation. The authors report phase-specific effect of optogenetic cholinergic modulation in the appetitive Y-maze long-term memory task, as well as the switches of the CA3 and CA1 activity from ripple activity to theta/gamma oscillations. This work builds on the previously published studies (Vandecasteele et al., 2014; Ma et al., 2020) that have already reported reduction in ripple incidence upon MS cholinergic stimulation. The novelty of this study lies in reporting the differences on learning dependent on the time (and location) of the simulation and apparent differences on the CA1 and CA3 ripples incidence. However, the conclusions in some cases are not strongly enough supported by the data. In my view several points need clarification and in particular the statistical underpowering resulting from very low number of animals in the results presented in Figure 5 speak against publication of the manuscript in the current form.1. The results showing decrease in learning following the MS cholinergic stimulation (2C-E) in the goal zones are convincing. However, it is known that SWRs may differ substantially in length, frequency and amplitude dependent on the behavioural state of the animal (ex. Joo and Frank, Nat Neuro Review 2019), which may have specific relevance for memory. Thus, by stimulating both the rewarded and the non-rewarded goal zones the effects of the stimulation on SWR may have differential effects on SWRs in rewarded and non-rewarded locations. The authors should compare SWRs and the effects of stimulation in rewarded and non-rewarded goal zones. A question is whether effects on SWRs in the non-rewarded goal zone are similar or substantially different, in which case the interpretation of how stimulation may have mechanistically contributed to memory formation could be fundamentally different.

We delivered light-stimulation in the goal area, in either the rewarded or the non-rewarded arm. The rationale for stimulating in both arms was to apply a similar duration of stimulation to all mice, irrespective of the progression of learning. We agree with the reviewer that SWRs occurring after rewarded and unrewarded trials likely differ. To give an overview of SWRs in the unrewarded goal zone, we now quantified and included data showing that a significantly smaller fraction of unrewarded than rewarded trials had one or more ripples in the goal zone (Figure 4D). Ripples in the unrewarded trials were likely less frequent than in the rewarded arm because mice spent less time in the arm (6.5 ± 0.5 s compared to 34.0 ± 1.0 s on rewarded trials), they were immobile for shorter periods, and because the reward promotes SWR activity (Singer and Frank 2009). We only recorded 21 ripples in the unrewarded non-stimulated trials that we could compare with the 171 ripples in the non-stimulated rewarded trials, but we do not think this could provide a conclusive comparison.

However, we do not attempt to disambiguate the role of rewarded and unrewarded trials in learning of the task. It is reasonable to assume that visits to the rewarded arm contributed to the memory of the rewarded location. By showing that SWRs were reduced by optogenetic stimulation at that stage of the task, we support the claim that reducing SWRs at that stage was relevant for the correlated memory impairment.

The Result section has been amended accordingly and now reads (page 12-13):

“We detected SWRs in significantly more rewarded than unrewarded trials (82 ± 7% of rewarded non-stimulated trials vs 32 ± 13% of unrewarded non-stimulated trials, paired t-test on percentages per animal: p = 0.02, n = 7 animals, Figure 4C). The difference could be due to the shorter immobility when the mice visited the non-rewarded arms: on unrewarded trials, mice spent 6.5 ± 0.5 s in the goal zone before leaving compared to 34.0 ± 1.0 s on rewarded trials. Because we detected few SWRs in the unrewarded trials, we restricted the statistical analysis of the effects of optogenetic stimulation to rewarded trials.”

“Spectral peak frequency of SWRs was not affected by the stimulation (Frequency: 168 ± 2 Hz; linear mixed-effects model for non-stimulated trials, mouse group – laser interaction: F_(1, 3.6)_ = 0.02, p = 0.88, Figure 4 —figure supplement 2A), nor was the SWR duration (Duration: 37 ± 1 ms; log-linear mixed-effects model for non-stimulated trials, mouse group – laser interaction: F_(1, 148)_ = 0.1, p = 0.76, Figure 4—figure supplement 2B).”

2. The authors show the reduction of ripples incidence in CA1 and CA3 upon MS cholinergic stimulation in both in sleeping and anesthetized animals. However, they also claim that "…MS cholinergic stimulation enhanced a scopolamine-sensitive theta oscillation in both anesthetized and sleeping mice (supplementary Figure 3)". To substantiate this claim by data, they need to block muscarinic receptors in sleeping animals to assure that it holds in both conditions. I did not find these data in the manuscript. Scopolamine sensitivity is a classical criterion for differentiation of types of theta oscillations so that this information would be highly relevant for the interpretation of the results with respect to effects of stimulation on theta oscillations. If the authors did not perform these experiments, they should not extend the interpretation to the sleep condition.

The Reviewer is correct – we are sorry for the inadvertent extrapolation from anesthetized to sleeping mice. In this majorly revised version of the manuscript, we have removed results showing the involvement of muscarinic receptors, which have been described extensively by others (Vandecasteele et al., 2014; Ma et al., 2020). We would argue that scopolamine injection to a non-anesthetized mouse, which could cause arrhythmia and have serious neurological effects, is probably not important enough to our study to be ethically justified.

3. Figure 4: Using the method presented in Haller et al., 2018, the authors show an increase in theta power and gamma power in both CA1 and CA3, following MS cholinergic stimulation. While these results are interesting, it would be much more convincing if the authors showed the grouped data of all animals, including the error bars in the Figure 4B, instead of only the data obtained from an individual animal. More importantly the statistical analysis depends on a linear fit of the background spectrum intensity, which from inspecting the data seems moderately imprecise. Since the differences reported are rather small, the authors should use a 1/f polynomial fit (as also used in Haller et al., 2018) of the background spectrum intensity which would confirm the robustness of the described differences.

We would argue that averaging spectra from different animals can blur variability in the PSD as the oscillatory frequency bands differ between individual mice (Haller et al. 2018, now out in Nat. Neuroscience as Donoghue et al., 2020). Therefore, we present PSDs from each animal individually, a representative one in Figure 3C and Figure 6B, and the others in the extended Figure 3—figure supplement 1 and Figure 6—figure supplement 1. To summarize the results, in the previous version of the manuscript, Figure 6B showed the mean PSD per animal and grey ribbons extending ±1 SEM. In some cases, the ribbons were narrower than the width of the mean line, so we updated the figures to only include the ribbons (Figure 4B, Figure 4—figure supplement 1, Figure 6B, Figure 6—figure supplement 1).

The Reviewer is correct to point out that fitting the background spectrum is imprecise. Following the Reviewer’s suggestions, we compared the fitting of linear and polynomial models for the background spectrum. The polynomial fit in 76 % of cases failed to detect relative theta peak in control trials. This is because the fitted polynomial background bent around the theta peak without any gaussian peaks added on top of the background. In comparison, 99 % of the same trials had a relative theta peak detected when fit with the linear background spectrum. Please Author response image 1 demonstrating the fits (shown with the dashed line) calculated on an animal-averaged PSD.

To independently confirm the increase in relative theta and slow gamma power in sleeping mice, we looked at the change in PSD between subsequent epochs with the stimulation off and on. For both theta and slow gamma, the negative power change was smaller than in the surrounding frequency bands. These significance-tested comparisons are presented in Figure 6C and Figure 6—figure supplement 2. The changes in the main text read (page 16):“To independently confirm that the stimulation increased relative theta power, we looked at the difference in the PSD between subsequent epochs with the stimulation off and on (Figure 6D, Figure 6—figure supplement 1). In the ChAT-ChR2 mice, the negative change in the theta band was significantly lower than in the 12–15 Hz band (mixed-effects model: F_(1, 10)_ = 21, p = 0.001, Figure 6—figure supplement 2A).”

“We independently confirmed the increase in relative slow gamma power by looking at the PSD change between subsequent epochs with the stimulation off and on. In the ChAT-ChR2 mice, the negative change of power in the slow gamma band was significantly lower than in the 12–15 Hz band (mouse group effect in the linear mixed-effects model: compared to the 12–15 Hz band: F_(1, 8)_ = 35, p = 10^-4^; compared to the 90–110 Hz band: F_(1, 10)_ = 3.7, p = 0.08, Figure 6—figure supplement 2B).”

4. Related to Figure 4: The authors state: "The increase in the power was associated with significantly lower frequency of the theta peak in the CA1, but not in the CA3." It would be helpful for the reader to also show these results in the Figure.

As suggested, we included these results for CA1 in Figure 6F. The text on page 16 reads:

“…spectral peak frequency in the theta band decreased from 7.7 ± 0.2 Hz to 7.2 ± 0.1 Hz (log-linear mixed-effects model, mouse group – laser interaction: F_(1, 4.8)_ = 7.3, p = 0.04, Figure 6F, post hoc test: t_(30)_ = 4.5, p = 0.001).”

5. The authors are reporting: "Slow gamma (25-45 Hz) increased in CA1 by 54 {plus minus} 6 % and in CA3 by 8 {plus minus} 8 % ". It has been previously shown that slow gamma is driven by CA3 and propagates to CA1 (Colgin et al., 2014, 2016, Schomburg et al., 2014). To my understanding, this would imply that the low gamma activity should reach CA1 with a smaller amplitude than measured in CA3. As the opposite is observed here, does that imply that the local CA1 network activity lead to amplification of the gamma power in CA1? This should be discussed.

We now removed the CA3 data from the manuscript, so this question is not relevant to the current version. We did not draw any conclusions about the relative differences in slow gamma oscillatory power between the CA1 and CA3 because there is no simple relation between the LFP and network activity. Just for interest, it is entirely possible that inhibitory currents are responsible for the major component of the slow gamma LFP in the CA3, while slow gamma is mediated primarily by excitatory currents in the CA1.

6. In the Figure 5, the authors report decreases in CA1, but not in CA3 ripples incidence in the goal zone following the MS cholinergic stimulation. I have several difficulties with the interpretation of the results presented in this Figure. First, the authors state that only successful trials were analysed. As mentioned above already, given that the stimulation was performed also during the unsuccessful trials, the results obtained in the unrewarded goal zones should be presented. Furthermore, it is stated: "…n=3 mice included in the analysis with the ripple incidence at goal {greater than or equal to} 0.03". I think that "{less than or equal to} 0.03" is meant here. In any case, looking carefully at the Figure 5D, a decrease of the ripple incidence at goal can be observed in 2 of the 3 animals while the increase observed in one single animal accounts for no difference which underlies the conclusion that the effects are specific for CA1. This experiment is strongly underpowered and the conclusions are not convincingly supported by the data.

In response to this concern, we conducted Y-maze experiments with two additional ChAT-ChR2 and two ChAT-GFP animals. The statistical comparison with the linear-mixed effects model was performed on samples from all trials. This accounts for 111 non-stimulated and 109 stimulated at goal rewarded trials. We changed the presentation in Figure 4E and other figures to display value from each trial used by the statistical test. We believe the increased number of trials, the comparison with the effect in the control mice, and the changed presentation strongly support the conclusion.

Regarding the inclusion of unrewarded trials, please see our response to the Reviewer’s point 1. Data for the CA3 is not included in the revised version of the manuscript.

7. Furthermore, no difference in theta and gamma power in CA1, but increase in slow gamma in CA3 (5E-5G) are reported. Looking carefully at 5I, a trend in increase could be also observed in CA1. Also here, the number of animals must be increased.

The increased sample size did not reject the null hypothesis of no effect of the stimulation on theta-gamma. We would like to emphasize that the statistical model was built using all of 111 non-stimulated and 109 stimulated goal trials recorded from 5 ChAT-ChR2 and 2 ChAT-GFP mice.

Data for the CA3 is not included in the revised version of the manuscript.

8. The title of the paper is: "Cholinergic suppression of sharp wave-ripples impairs hippocampus dependent spatial memory" The authors are also showing the effect of MS cholinergic stimulation on the SWRs during sleep, but the link between this effect and spatial memory is not explored. It is known that supressing SWRs specifically during the post-learning sleep impairs spatial memory (Girardeau et al., Nature Neuroscience 2009). Thus, it would be interesting to see whether the MS cholinergic stimulation and SWRs suppression during sleep impairs spatial memory. This should be at least discussed.

We agree with the Reviewer that it would be of interest to see how this impairment relates to suppressed SWRs during post-learning sleep but we think this question is out of the scope for the present paper. We added the following paragraph about the relation to post-learning SWRs in the Discussion (page 20):

“Because learning can be affected by the interruption of SWRs during post-learning sleep (Girardeau et al., 2004), and because our cholinergic activation during sleep achieves a similar effect on the SWRs (Figure 3; Ma et al., 2020), it would be of interest to see if the cholinergic activation during post-learning sleep would also impair spatial learning. This would show whether low-cholinergic states are important also for memory consolidation during sleep and provide further evidence for a possible role of SWRs in memory.”

Reviewer #3:The study focused on the effect of MS cholinergic activity on hippocampal oscillations and extended our knowledge of cholinergic function in hippocampus-dependent spatial learning. Impairment of hippocampal ripple oscillations during awake immobility leads to significant performance deficit in the Y-maze task, highlighting the importance of precise timing of cholinergic input in memory formation.1. The conclusion points 1/2/3 in the abstract maybe more coherent if rearranged to 3/1/2. But this might lead to structural re-organization of the article and the figures (suggested figure order: Figure 1-3-4-2-5). The logic behind this order is:a. Optogenetic stimulation of MS cholinergic neurons impair hippocampal ripples during SWS (Figure 3/4).b. What about ripples during awake immobility?c. Optogenetic stimulation during goal period of the Y-maze impairs learning (Figure 2).d. Such impairment was due to reduced CA1 ripple incidence (Figure 5).

We thank the Reviewer for this suggestion. We have now reorganized the manuscript and order of figures to mirror the conclusion points in the abstract, which is slightly different to the Reviewer’s preference. We prefer to present up front our main finding, which is that the timing of cholinergic neuron activation is crucial for learning and memory (Figure 2). Then we proceed with the recordings and optogenetics during the memory task and end with the sleep recordings, motivated by our failure to detect a significant effect of cholinergic stimulation on theta-gamma activity. Please also see response to Reviewer 1, point 5. Unless the Reviewer considers the order of the figures to be critical for the article to be acceptable, we would rather keep the current organization.

2. In the introduction: "activation of MS cholinergic neurons switches the CA3 and CA1 network states from ripple activity to theta/gamma oscillations". The phrasing might be questionable

We assume that the Reviewer questions the clear distinction we make between the neural states: ripple activity state and theta/gamma state. We have rephrased the sentence, which now reads (page 6):

“We also show that activation of MS cholinergic neurons promotes a switch from ripple activity to enhanced theta/gamma oscillations in the hippocampus of naturally sleeping mice.”

3. Page 18, first paragraph, the description seems a little bit redundant.

We agree with the Reviewer that this paragraph reiterates the already described findings. To avoid redundancy, we have now clearly separated this Result section into five paragraphs: (1) description of the task and of the mouse cohorts; (2) description of the main results; (3) influence of stimulation duration (as recommended by Reviewer 1); (4) control of potential aversive influence of light at the goal location; (5) conclusion of the section.

4. Figure 3A, it seems that theta oscillations emerge with the optogenetic stimulation, was the stimulation strictly delivered during SWS or was this particular theta represents REM states? It would be better if the authors also show the LFP and ripple trace after the light stimulation.

We delivered the stimulation with no distinction between SWS and REM sleep. We added this information to the manuscript and included the requested trace after stimulation in Figure 6A. It now reads (page 14):

“We compared the signal in the 30 s-long epochs preceding the stimulation with the 30-s-long epochs during the stimulation without a distinction between SWS and REM sleep.”

5. Page 22, last sentence of the first paragraph, "…while the peak frequency did not significantly change from 39 {plus minus} 1 Hz in the CA1 and 38 {plus minus} 2 Hz in the CA3…", is a little bit confusing, needs further explanation.

By peak frequency, we mean frequency with spectral peak power. We updated the references to spectral peak frequency throughout the manuscript.

6. Page 22, second paragraph, relative theta and gamma power change was reported in CA3, how about CA1?

Because these results have been published by two groups already (Vandecasteele et al., 2015; Ma et al., 2020), we have now removed the anesthetized data from the manuscript and focus on natural sleep and behaving mice.

7. In previous reports (Vandecasteele, 2014 and Ma, 2020), MS cholinergic stimulation can completely inhibit hippocampal ripple oscillations. But in this study, there are still a lot of ripples not suppressed. Please explain the difference.

We still observed some SWRs during MS cholinergic stimulation, but in similar proportions to those published elsewhere. We report that SWR incidence in 4 ChAT-ChR2 decreased by 100% and for another 4 reduced by a median of 88 ± 2%. These are comparable to both of the other reports:

1. In freely behaving animals, Vandecasteele et al. (2014) report a median 92% reduction in SWRs. The relevant result quoted from Vandecasteele et al:

“Ripple occurrence was significantly suppressed or abolished during MS stimulation (1–12 Hz, sine stimulation or pulse trains, 1–60 s) in mice recorded either during urethane anesthesia (n = 6 mice, median suppression −90%, P = 0.0312) or during free behavior (n = 8, median suppression −92%, P < 0.01).”

2. Ma et al. (2020) report 95 ± 5% reduction in SWRs (reported as ripple events inhibition index).

[Editors’ note: what follows is the authors’ response to the second round of review.]

Essential Revisions:Some conclusions that are strongly stated in the current version will need supporting data in the future or will need to be toned down with caveats discussed. Please refer to the individual reviews below for specific details.

We agree. We have toned down our conclusions accordingly and included a discussion of caveats as described in our responses to the individual Reviewers.

Reviewer #1 (Recommendations for the authors):The authors satisfactorily addressed my prior major concerns. I have no major concerns remaining. I only have a few concerns remaining that can be easily addressed by the authors without affecting the major conclusions of the paper.Page 5: "In hippocampal CA3, cholinergic activation induces a slow gamma rhythm primarily by activating M1 muscarinic receptors": It should be noted that these are in vitro studies. As this sentence is written currently, the comparisons between CA3 and CA1 are potentially misleading.

We thank the Reviewer for pointing out this potentially misleading sentence. We now explicitly state that the CA3 results were obtained ex vivo. The sentence now reads (page 5):

“In hippocampal CA3, cholinergic activation ex vivo induces a slow gamma rhythm primarily by activating M1 muscarinic receptors (Fisahn et al., 1998; Betterton et al., 2017), while in the CA1, cholinergic activation in vivo promotes theta/gamma oscillations…”

Page 10: "We did not detect an effect of the laser on the duration the mice spent at the goal location (linear mixed-effects model, mouse group – laser interaction: F(1, 78) = 0.01, p = 0.94, laser effect: F(1, 78) = 0.1, p = 0.73)." I am confused by this passage. If optogenetic stimulation at the reward location affects memory, as the authors claim, one would expect a differential effect of laser stimulation on the ChR2 mice compared to the GFP-only control mice (and a significant effect of the laser in the ChR2 group). Yet, a non-significant interaction effect is reported.

The comparison looks at the time the mice spent at the goal location, which is not a measure of performance in the task but assesses the behavior during learning. Thus, in this passage, we compared the duration the GFP-only and ChR2-expressing mice spent at the goal locations, and not the performance of the task. The cholinergic stimulation did not result in overt behavioral changes, for example, in changes that could indicate the stimulation itself was aversive. We apologize for the confusion and have now rephrased the sentence to make the comparison more explicit.

We have amended the sentence as follows (page 10):

“The cholinergic activation did not overtly affect the behavior once the mice were at the goal location: we did not detect any effect of the laser stimulation on the time the mice spent at the goal location (linear mixed-effects model, mouse group – laser interaction: F(1, 78) = 0.01, p = 0.94, laser effect: F(1, 78) = 0.1, p = 0.73).”

Page 13: "Because we detected few SWRs in the unrewarded trials, we restricted the statistical analysis of the effects of optogenetic stimulation to rewarded trials. The SWR incidence in the non-stimulated trials was not significantly different between early (before day 5) and late learning (linear mixed-effects model, effect of early vs late learning: F(1, 110) = 0.3, p = 0.58, Figure 4D).": The authors state that they restricted analysis of effects of optogenetic stimulation to rewarded trials but then analyze SWR incidence in the very next sentence and paragraph. Is there a typo here?

We have now clarified that the statistical comparison relates to SWR incidence during early and late learning in non-stimulated rewarded trials (page 13):

“Because we detected few SWRs in the unrewarded trials, we restricted the further analysis to the rewarded trials. We first assessed whether SWR incidence changed during learning by quantifying the incidence of SWRs in the non-stimulated rewarded trials during early and late learning (Figure 4D). We did not observe any significant difference between early (before day 5) and late learning (linear mixed-effects model, effect of early vs late learning: F(1, 110) = 0.3, p = 0.58, Figure 4D). “

Reviewer #2 (Recommendations for the authors):This submitted work is clearly better structured and well-controlled version of the previously submitted manuscript. The authors addressed most of the comments. I only have two concerns.1. The optogenetic stimulation induced ripple inhibition effect at the goal location and during sleep seems a little bit inconsistent (Figure 4E and Figure 5C). The authors stated in the text the ripple inhibition statistics under the two conditions: ripple incidence was reduced from 0.11 {plus minus} 0.01 Hz to 0.05 {plus minus} 0.01 Hz when stimulated at the goal location, and from 0.21 {plus minus} 0.01 Hz to 0.03 {plus minus} 0.01 Hz during sleep. Also, from the authors' response to the review comments (Reviewer #3, major comment 7), "We report that SWR incidence in 4 ChAT-ChR2 decreased by 100% and for another 4 reduced by a median of 88 {plus minus} 2%.". I assume that the authors were referring to the situation during sleep (based on the statistics and the number of animals used in each condition), which means that the ripple inhibition effect would be somewhere around 50% during stimulation at the goal location. Please elaborate on this.

The Reviewer is correct. We apologize for the ambiguity in our previous response to Reviewer 3. As requested, in the Results section, we now present the effect sizes as percentage change along with the incidence of SWRs in control and during optogenetic stimulation. In the revised Discussion section, we elaborate on the lower efficacy of optogenetic stimulation on reduction of SWR incidence in the awake behaving animal compared to during sleep.

In addition, when we recalculated the mean statistics, we found a mistake in the reported mean SWR incidence at the goal location. In the optogenetically stimulated trials, the incidence decreased by 52 ± 7% from 0.06 ± 0.01 Hz to 0.03 ± 0.01 Hz.

We have amended the main text to include the percentages and have updated the statistics. The Results section now reads:

Page 13 reporting the reduction at goal location:

“…optogenetic stimulation had a significantly different effect in the ChAT-GFP and the ChAT-ChR2 mice (log-linear mixed-effects model, mouse group – laser interaction, F(1,42) = 4.5, p = 0.04, Figure 4E). In the ChAT-ChR2 mice, optogenetic stimulation reduced the SWR incidence at the goal location by 52 ± 7% from 0.06 ± 0.01 Hz to 0.03 ± 0.01 Hz (post hoc test: t(44) = 4.2, p = 0.001, Figure 4E).”

Page 14 reporting the mean ± SEM reduction at goal location: “Also, the reduction of SWR incidence of 52 ± 7% at the goal location was smaller than the 92% median suppression reported during free behavior (Vandecasteele et al., 2014), which could be due to a smaller effect of ACh at the reward location or an already high level occluding the effect of the optogenetic stimulation.”

Page 15 reporting the mean ± SEM reduction in sleeping animals: “Optogenetic stimulation reduced the SWR incidence throughout the stimulation in ChAT-ChR2 mice but not in ChAT-GFP mice (Figure 5A–C). SWR incidence in ChAT-ChR2 mice was reduced from 0.21 ± 0.01 Hz to 0.03 ± 0.01 Hz (85 ± 3% reduction), linear mixed-effects model, mouse group – laser interaction: F(1, 22) = 47, p = 10-6, n = 369 epochs from 10 animals…”

Page 18: “Our results indicate that cholinergic stimulation almost completely suppresses SWRs in sleeping animals and suppresses SWRs by about one half in awake, behaving animals.”

2. Although the analyses of the effect of optogenetic stimulation on hippocampal theta and gamma oscillations during sleep reveal some very interesting results, it seems that the relevance of these analyses with the key claim of the submitted work (as indicated by the title) is a little bit farfetched.

We agree with the Reviewer that the sleep recordings are not directly linked to the main results of the study. However, we think that these results are valuable for the main story and help us understand the effect of stimulating the cholinergic neurons during the Y-maze task for two main reasons.

First, the incomplete suppression of SWRs during behavior as well as the weak effect on theta and gamma oscillations could be attributed to a deficient stimulation paradigm. However, our results during sleep matches previously published results and gave us an appropriate opportunity to use the method developed by Donoghue and collaborators. Our results during slow-wave sleep, when the cholinergic tone is known to be low, highlights the powerful effect of cholinergic stimulation. In comparison, in the behaving mouse, when the cholinergic tone is higher, the stimulation has a more modest effect on the oscillations, but still results in behavioral impairment. Without recordings in the sleeping animals we would not have been able to draw such conclusions.

Second, we found it surprising that the cholinergic activation had no detectable impact on the theta-gamma oscillations at the goal location. This shows the prominent effect on theta-gamma in the sleeping animals is not seen in awake behaving animals and therefore is not likely to explain the learning impairment seen when cholinergic neurons were activated at the goal location.

We now highlight the second point explicitly in the discussion on page 18:

“Moreover, the effect of cholinergic stimulation on theta-gamma oscillations, which was prominent during sleep, was not observed when we applied the same stimulation at the goal location during learning, suggesting that learning was impaired through a mechanism independent of theta-gamma oscillations.”

Reviewer #3 (Recommendations for the authors):1. Figure 3B: the spectrogram shows high relative (z-scored) power for the high frequencies in start and center, but not goal. I would have expected to see at least some high power in the ripple band in the goal. Could the authors clarify how exactly the z-scoring was performed? If the spectrogram is an average across multiple trials, then this will tend to obscure transient, non-time locked oscillations like SWRs.

The Reviewer is correct to point out that averaging across multiple trials obscured transient SWRs from the spectrogram. To demonstrate these, we replaced the previous day-averaged spectrogram with a single trial spectrogram example in Figure 3B. The example shows transient power increases in the ripple frequencies at the goal location. The high-frequencies power also increased during running at Center for 60–75 Hz and harmonics of this frequency band (120–150 Hz and 180–225 Hz).

2. What is the effect of optical stimulation of cholinergic neurons on theta/gamma/SWRs in the "throughout" and "navigation" conditions? Are these effects consistent with the hypothesis that the learning deficit is caused by a reduction of SWRs at the goal location? Could additional insights be obtained into possible changes induced by stimulation (e.g. theta oscillations during navigation) that do *not* correlate with the learning deficit?

Although we agree with the Reviewer that these comparisons would be informative, unfortunately, we did not perform recordings in the ‘navigation’ and ‘throughout’ groups.

3. Page 19: the authors cite Jadhav et al. 2012 when stating "disruption of SWRs in the first 15 to 60 minutes following training impairs learning of spatial navigation tasks". However, Jadhav et al. disrupted SWRs during the training and not following the training.

The Reviewer is correct. This was a mistake, for which we apologize. We have now removed the citation from that sentence.

4. Page 20: Both reverse and forward replay are observed during brief pauses or reward consumption in the awake state when animals explore a maze or learn a task. So, it is likely that in the reward zone in the Y-maze task one will observe both forward and reverse replay. While it is fine to speculate that disruption of reverse replay mediates the behavioral deficit, it cannot be based on the assumption that replay at the goal location is only of the reverse kind.

We corrected the Discussion to reflect that replay events in both directions could occur at the goal locations (page 20):

“During these SWRs, sequences of neuronal activation are replayed in both forward and reverse order (Foster and Wilson, 2006; Csicsvari et al., 2007; Diba and Buzsáki, 2007; Karlsson and Frank, 2009; Ambrose et al., 2016).”

We removed the speculation about the direction of the ripple events (page 20):

“Therefore, we speculate that disruption of the normally occurring replay events in the reward zone is sufficient to impair long-term memory formation (Figure 5).”

5. What is the time in between individual trials?

We added this information in the Methods section: “The interval between the within-day trials averaged 10 minutes.“

6. To characterize the learning in the Y-maze, the authors determine the day at which criterion is reached. This metric is rather coarse. Instead, the authors could fit a learning curve (e.g. sigmoid function) to the trial responses and estimate the learning rate for each animal. Furthermore, it would be informative to show individual learning curves for all animals, in addition to the average learning curves that are shown now.

We agree with the Reviewer that the days-to-criterion metric is coarse. However, we do not think we can convincingly fit sigmoids to the individual learning trajectories in this task for two reasons:

1. Mice learnt the task fast, for example, 10 mice reached the learning criterion in one or two days, which leaves too few data points for adequate curve fitting.

2. Day-to-day progression was variable and the learning often was not gradual because the mice received only 10 tests a day. For example, some mice with performance of 70-90% on the first learning day had low performance on the second day.

Following the Reviewer’s suggestion, we now show individual learning curves in Figure 2—figure supplement 1.

7. To assess the effect of stimulation at the goal on hippocampal activity, the authors look at average SWR rate and average theta/gamma power. However, when the animals are in the goal region, they likely show a mixture of behavioral states that is associated with periods of theta and non-theta (incl. SWRs). Is more (or less) time spent in theta state during stimulation? Could it be that time spent in non-theta states is lower, but SWR rate within this state has not changed? Judging from the example in figure 4B, it may be the case that with stimulation the first SWR after arriving at the goal is delayed compared to no stimulation condition – is this consistent across all subjects?

The Reviewer is correct to point out that both periods with theta and non-theta are present at the goal location and he poses an interesting question. Unfortunately, in our view, the distinction between theta and non-theta states is not as unambiguous as previous literature might appear to imply. Whilst theta activity can be clearly identified during locomotor activity during navigation on the arms of the maze, we do not think our recordings would allow us to unequivocally distinguish between the time spent in theta vs non-theta states at the reward location. Therefore, we would prefer reporting only what we can unambiguously measure, namely the overall power in different frequency bands and the SWR incidence.